# p21 maintains senescent cell viability under persistent DNA damage response by restraining JNK and caspase signaling

Reut Yosef[1], Noam Pilpel[1], Nurit Papismadov[1], Hilah Gal[1], Yossi Ovadya[1], Ezra Vadai[1], Stav Miller[1], Ziv Porat[2], Shifra Ben-Dor[2] (iD) & Valery Krizhanovsky[1,*] (iD)

## Abstract

Cellular senescence is a permanent state of cell cycle arrest that protects the organism from tumorigenesis and regulates tissue integrity upon damage and during tissue remodeling. However, accumulation of senescent cells in tissues during aging contributes to age-related pathologies. A deeper understanding of the mechanisms regulating the viability of senescent cells is therefore required. Here, we show that the CDK inhibitor p21 (CDKN1A) maintains the viability of DNA damage-induced senescent cells. Upon p21 knockdown, senescent cells acquired multiple DNA lesions that activated ataxia telangiectasia mutated (ATM) and nuclear factor (NF)-κB kinase, leading to decreased cell survival. NF-κB activation induced TNF-α secretion and JNK activation to mediate death of senescent cells in a caspase- and JNK-dependent manner. Notably, p21 knockout in mice eliminated liver senescent stellate cells and alleviated liver fibrosis and collagen production. These findings define a novel pathway that regulates senescent cell viability and fibrosis.

**Keywords** apoptosis; cellular senescence; DNA damage response; JNK; p21 (CDKN1A)
**Subject Categories** Cell Cycle; DNA Replication, Repair & Recombination; Molecular Biology of Disease
**The EMBO Journal (2017) 36: 2280–2295**

## Introduction

Cellular senescence, a permanent form of cell cycle arrest, halts cell proliferation in response to various stressors, including telomere shortening, oncogene activation, and DNA damage (Kuilman *et al*, 2010; Campisi, 2013; Munoz-Espin & Serrano, 2014; Salama *et al*, 2014). Senescent cells present elevated activity of senescence-associated beta-galactosidase (SA-β-gal) and a persistent DNA damage response that distinguish them from other non-proliferating cell populations (Campisi & d'Adda di Fagagna, 2007; Collado *et al*, 2007; d'Adda di Fagagna, 2008). In addition, senescent cells produce a variety of characteristic secreted factors, collectively termed the senescence-associated secretory phenotype (SASP) (Coppe *et al*, 2008), which reinforces senescence arrest in an autocrine manner (Acosta *et al*, 2008, 2013; Kuilman *et al*, 2008; Wajapeyee *et al*, 2008) and mediates immune surveillance of the senescent cells (Sagiv & Krizhanovsky, 2013; Sagiv *et al*, 2016). With aging, however, senescent cells accumulate in the organism promoting local inflammation that drives tissue aging, tissue destruction, and potentially also tumorigenesis and metastasis in a cell non-autonomous manner (Krtolica *et al*, 2001; Campisi, 2013; Munoz-Espin & Serrano, 2014; Ovadya & Krizhanovsky, 2014; Salama *et al*, 2014). Recent studies have shown that elimination of senescent cells promotes stem cell proliferation and prolongs lifespan (Baker *et al*, 2011, 2016; Chang *et al*, 2016; Yosef *et al*, 2016). Therefore, mechanisms that regulate the viability of senescent cells in tissues evidently play an important role in tissue homeostasis.

Cellular senescence indeed plays an important role in tissue homeostasis after short-term damage, facilitating tissue repair after liver damage, in skin fibrosis, and during wound healing (Krizhanovsky *et al*, 2008; Jun & Lau, 2010; Demaria *et al*, 2014). Upon liver damage, for example, hepatic stellate cells (HSCs) differentiate into activated myofibroblasts that extensively proliferate and secrete extracellular matrix (ECM). Accumulation of ECM leads to fibrosis, which aims to maintain tissue integrity while the healing process is taking place. These activated HSCs subsequently undergo senescence, accompanied by a decline in ECM production and an increase in SASP factors that facilitate elimination of these cells by natural killer (NK) cells (Krizhanovsky *et al*, 2008). The matricellular protein CCN1, which is secreted from damaged hepatocytes, can trigger senescence in activated HSCs (Kim *et al*, 2013). Inactivation of the senescence specifically in HSCs results in enhanced tissue fibrosis, indicating that senescence limits fibrosis (Krizhanovsky *et al*, 2008). However, inefficient elimination of senescent cells and their excessive accumulation are associated with increased fibrosis (Krizhanovsky *et al*, 2008; Sagiv *et al*, 2013, 2016).

1   Department of Molecular Cell Biology, The Weizmann Institute of Science, Rehovot, Israel
2   Life Sciences Core Facilities, The Weizmann Institute of Science, Rehovot, Israel
    *Corresponding author. Tel: +972 8 934 6575; Fax: +972 8 934 4125; E-mail: valery.krizhanovsky@weizmann.ac.il

 

The senescence program is driven by a complex interplay of signaling pathways. To promote and support cell cycle arrest, p16$^{INK4A}$ (CDKN2A), accompanied by the p53 (TP53) target p21 (CDKN1A), inhibits cyclin-dependent kinases (CDKs), thereby preventing phosphorylation of the retinoblastoma protein (pRb) and thus in turn suppressing the expression of proliferation-associated genes (Narita *et al*, 2003, 2006; Collado *et al*, 2007). In addition, the nuclear factor kappa B protein complex (NF-κB) acts as a master regulator of SASP and therefore affects both the microenvironment of senescent cells and their immune surveillance (Acosta *et al*, 2008; Krizhanovsky *et al*, 2008; Xue *et al*, 2011; Lasorella *et al*, 2014). Clearance of senescent cells by the immune system is an essential requirement for limiting their prolonged retention in tissues, a trait that might derive from their intrinsic resistance to apoptosis (Yosef *et al*, 2016).

Whereas mechanisms driving senescence have been extensively studied, the mechanisms allowing their prolonged retention in tissues are much less well characterized. Recently, the anti-apoptotic BCL-2 family members BCL-W, BCL-XL, and BCL-2 were shown to facilitate the resistance of senescent cells to apoptosis (Chang *et al*, 2016; Yosef *et al*, 2016). However, the contribution of pathways that regulate the formation of senescent cells to the resistance of these cells to cell death has yet to be determined. On one hand, senescent cells cannot accumulate p53 protein to the levels required for apoptosis (Seluanov *et al*, 2001). On the other hand, the p53 target p21, via its ability to promote cell cycle inhibition, can protect some cells from apoptosis (Abbas & Dutta, 2009). This effect might be governed by both p53-dependent and -independent upregulation of the pro-apoptotic protein BAX, or by activation of members of the tumor necrosis factor (TNF)-α family of death receptors, or by effects on DNA repair (Abbas & Dutta, 2009). We therefore set out to determine how p21 regulates the viability of senescent cells after DNA damage. We found that following p21 knockdown, senescent cells sustain multiple DNA lesions, leading to further activation of DNA damage response and NF-κB pathways. This activation was regulated by both TNF-α secretion and JNK activation, and it mediated senescent cell death in a caspase-dependent and JNK-dependent manner. Moreover, p21 knockout in mice led to the elimination of senescent cells from fibrotic scars in the liver and alleviated liver fibrosis. These results uncovered new mechanisms that control the fate of senescent cells.

# Results

## p21 maintains the viability of DNA damage-induced senescent cells

p21 contributes to the induction of cellular senescence owing to its activity as an inhibitor of CDK2 and CDK4 (Campisi & d'Adda di Fagagna, 2007). To examine the contribution of p21 to the viability of senescent cells, we induced senescence in normal human lung fibroblasts (IMR-90) by treating them with the DNA-damaging agent etoposide ("DNA damage-induced senescence", DIS). We confirmed the induction of the senescence phenotype in these cells by evaluation of cell cycle arrest, expression of senescence markers p16, p21, and p53 and senescence-associated β-galactosidase (SA-β-gal) staining (Appendix Fig S1A–E). Oncogene-induced senescence (OIS) was induced by expression of oncogenic H-Ras$^{V12}$. Proliferating (growing)

cells and empty vector-transduced cells served as controls for DIS and OIS cells, respectively. Senescent and control IMR-90 cells were treated with siRNA mixes targeting p21 (sip21) or control siRNAs (siCtrl). The efficiency of the knockdown was confirmed by Western blot analysis (Fig 1A). Surprisingly, knockdown of p21 induced death in 30% of the DIS cells only (Fig 1A). The viability of control or of OIS cells was not affected by p21 knockdown. To determine whether this unexpected observation was limited to a specific cell type, we performed p21 knockdown in mouse embryonic fibroblasts (MEFs) and in normal human skin fibroblasts (BJ cells). Growing (G) and DIS MEFs or BJ cells were treated with siRNA mixes targeting p21, and their viabilities were evaluated. Knockdown of p21 reduced the viability of DIS MEFs by 26% relative to siCtrl cells (Fig 1B). p21 knockdown decreased the viability DIS BJ cells even more strongly, leading to a 54% reduction in sip21 relative to siCtrl-treated DIS cells (Fig 1C). Similarly, p21 knockdown decreases the viability of DIS cells 3 weeks following the induction of DIS phenotype (Appendix Fig S1F). Thus, these results indicate that p21 is necessary to maintain the viability of DIS cells, regardless of the time passed from the induction of senescence.

To find out whether the induction of cell death is dependent on the time of p21 knockdown, we analyzed cells that were already lacking p21 prior to introduction of the DNA-damaging agent. To this end, we induced senescence in wild-type (WT) and p21 knockout (p21$^{-/-}$) MEFs. After exposure to DNA-damaging agent leading to senescence induction, the viability of p21$^{-/-}$ MEFs was decreased by 60% relative to WT MEFs (Fig 1D). Therefore, p21 supports viability of cells regardless the timing of the knockdown.

Cancer cells can acquire senescence-like phenotypes in response to DNA damage (Appendix Fig S2A and B). To impose this phenotype, we transduced H1299 cells with small hairpin RNA (shRNA) targeting p21 (shp21) or control shRNA targeting Luciferase (shLuci) and then treated the cells with etoposide to induce DNA damage. Treatment with etoposide induced cell cycle arrest in these cells (Appendix Fig S2C). Knockdown of p21 in this setting caused a 75% reduction in the viability of etoposide-treated cells relative to shLuci cells (Appendix Fig S2D). Thus, the effect of p21 knockdown on the viability of cells after damage to their DNA is not limited to normal fibroblasts.

To determine the time at which cell death occurs after p21 knockdown, we monitored cell viability over time course following knockdown. Importantly, p21 knockdown was followed by continuous reduction in DIS BJ cell viability relative to control cells over time (Fig 1E). These results suggest that the effect of p21 knockdown on DIS cell viability is cumulative.

## Molecular pathways activated after p21 knockdown in DIS cells

To identify the molecular mechanism controlling DIS cell viability, we studied the expression patterns of DIS and control cells with and without p21 knockdown. Growing and DIS BJ cells were transfected with siRNAs against p21 or with control siRNAs. After 3 days, total RNA was extracted and gene expression was determined using Affymetrix microarrays. K-means clustering (Fig 2A) and principal component analysis (PCA; Fig 2B) were used to visualize the overall response to p21 knockdown. A massive change in gene-expression profile was detected after p21 knockdown in DIS cells but not in the growing control cells. Overall, the signal intensity of 1,595 unique genes changed significantly in response to p21 knockdown in DIS

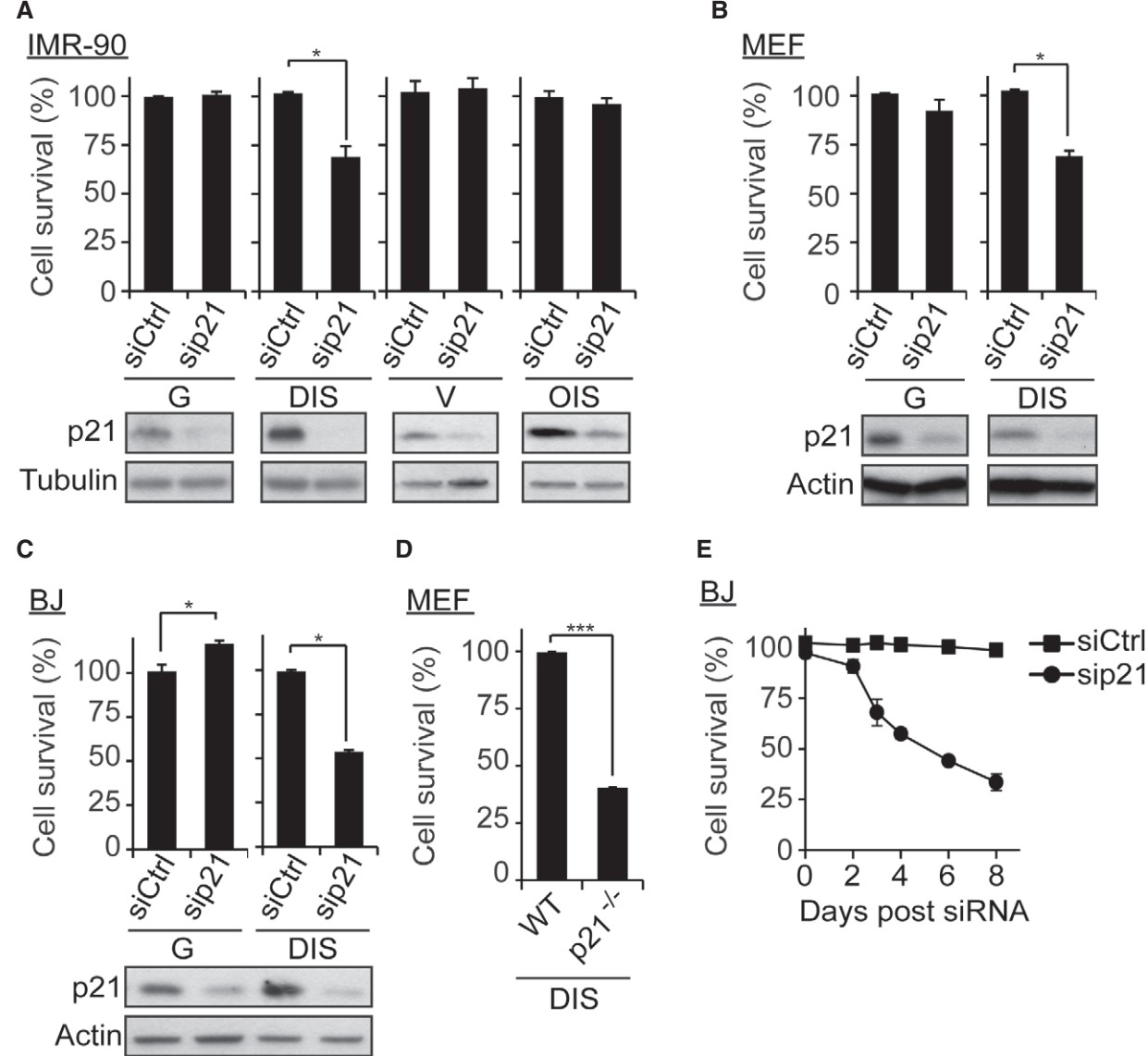

**Figure 1.  p21 maintains the viability of DNA damage-induced senescent cells.**

A   Normal human IMR-90 fibroblasts were induced to senesce by DNA damage (DNA damage-induced senescent (DIS)) or by expression of Hras$^{V12}$ (oncogene-induced senescent (OIS)). Proliferating IMR-90 (growing, G) and empty vector-transfected (V) cells served as respective controls. The cells were transduced with siRNAs targeting p21 (sip21) or control siRNAs (siCtrl). Histograms indicate the percentages of surviving senescent cells compared with G or V controls. Western blots show p21 protein levels in the corresponding samples.

B   Mouse embryonic fibroblasts (MEF; G and DIS) were transduced with siRNAs targeting p21 or control siRNA, and cell survival was evaluated. Western blots show p21 protein levels in the corresponding samples.

C   BJ human fibroblasts (DIS and G cells) were transduced with siRNAs targeting p21 or control siRNA, and cell survival was determined 4 days following the transduction. Western blots show p21 protein levels in the corresponding samples.

D   Mouse embryonic fibroblasts (MEF) from wild-type (WT) or p21 knockout mice (p21$^{-/-}$) were treated with etoposide to induce DNA damage-induced senescence. Survival of DIS cells was determined by quantification of the remaining adherent cells 5 days following etoposide washout.

E   BJ DIS fibroblasts were transduced with siRNAs targeting p21 and cell survival was determined at the indicated time points.

Data information: Data are presented as mean ± SEM of three repeats, performed in triplicates. Data were analyzed using Student's *t*-test. *$P < 0.05$, ***$P < 0.0005$.

cells compared to only 82 in growing cells (Fig 2C). Therefore, p21 knockdown in DIS cells induces widespread albeit specific changes in gene expression.

Cluster and principal component analyses describe overall changes in gene expression. To further identify the perturbed biological networks and understand the biological processes and signaling networks invoked when p21 is silenced, we analyzed the data by enrichment analysis for pathways in the WikiPathways database (Kutmon *et al*, 2015) and by Ingenuity Pathway Analysis (IPA®, QIAGEN Redwood City, www.qiagen.com/ingenuity).

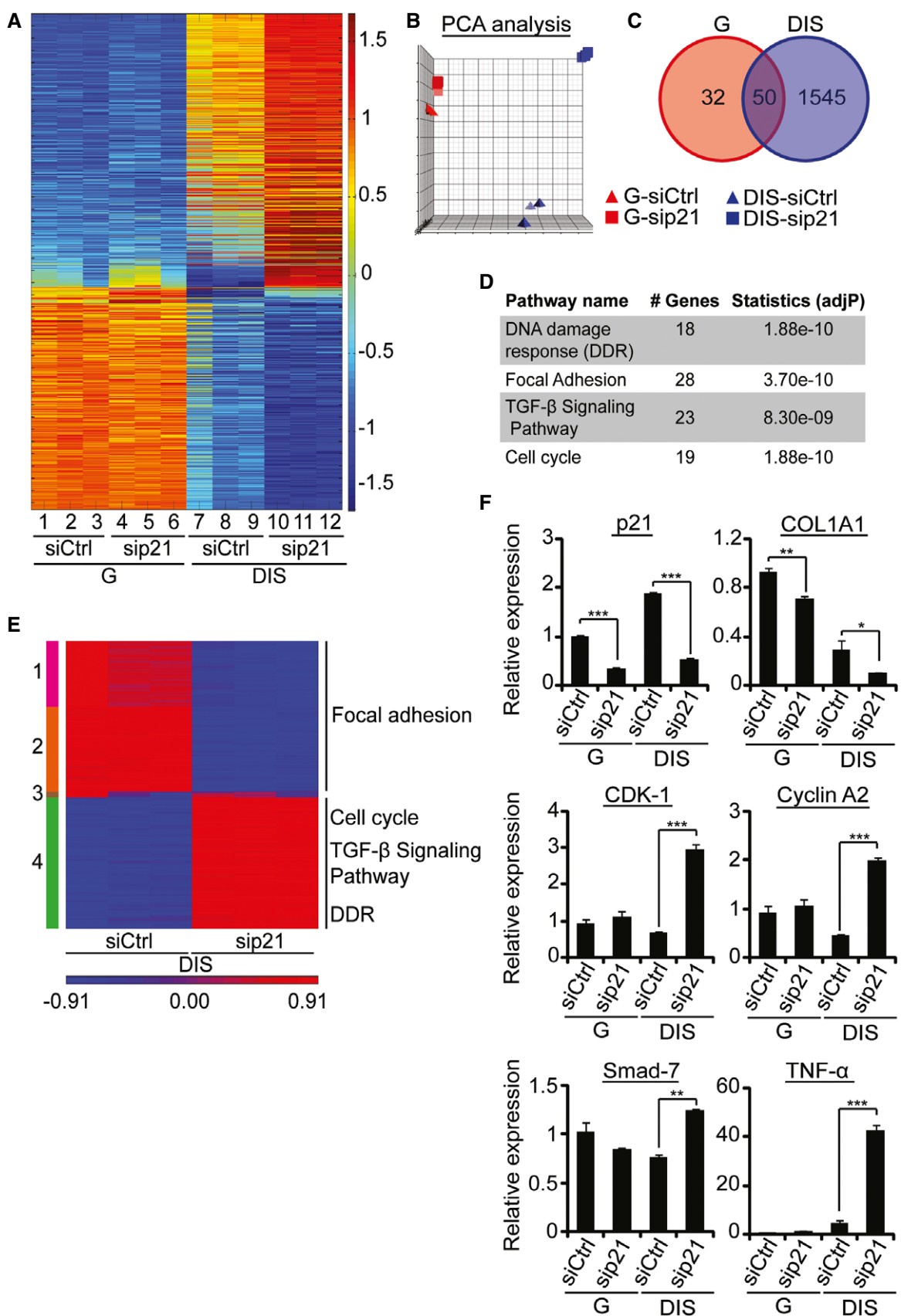

Figure 2.

Figure 2.   Gene-expression profiles of growing and senescent BJ cells after p21 knockdown.
BJ human fibroblasts (proliferating, G; and DNA damage-induced senescent, DIS) were transduced with either siRNA targeting p21 (sip21) or control siRNA (siCtrl). Cells were harvested and analyzed by Affymetrix PrimeView microarrays (3 replicates).

A   Results are presented as K-means clustering of the microarray data. Probe sets whose abundance was above the mean are shown in red, those below the mean in blue, and those equivalent to the mean in green.
B   Principal component analysis (PCA) scatterplot. Points are colored according to cell type (G, red; DIS, blue). Squares and triangles are drawn for sip21 and siCtrl siRNA groups, respectively.
C   Venn diagram showing the distribution of shared genes among G and DIS cells after p21 knockdown.
D   Enrichment analysis from the WikiPathways database identified pathways affected in 1,545 genes that were uniquely changed in DIS cells after p21 knockdown.
E   K-means clustering of the 1,545 genes that were uniquely changed in DIS BJ cells after p21 knockdown. Probe sets whose abundance was above the mean are shown in red, and those below the mean in blue.
F   mRNA expression levels relative to controls of p21, COL1A1, CDK-1, cyclin A2, Smad-7, and TNF-α genes after transduction with sip21 or control siRNA in G and DIS BJ cells. Data are presented as means ± SEM of three repeats, each performed in triplicate. Data were analyzed using Student's *t*-test. *$P < 0.05$, **$P < 0.005$, ***$P < 0.0005$.

WikiPathways analysis combined with K-means clustering indicated that the cell cycle regulators, transforming growth factor-β (TGF-β), the DNA damage response, and focal adhesion signaling pathways were altered in response to p21 knockdown in DIS cells (Fig 2D and E). Lists of the genes altered in each pathway, as well as pathway schemes, can be found in the Appendix (Appendix Figs S3–S6; Appendix Tables S1–S4). These findings were further supported by Ingenuity® Pathway Analysis (data not shown), which also predicted that p21 silencing would affect liver fibrosis via modulation of early signaling pathways in HSCs, mainly by modulation of collagen production and TGF-β signaling (Appendix Fig S7). To verify the microarray results, we performed RT–PCR analysis on specific genes in the pathways identified by the data analyses. Knockdown of p21 led to reduced expression of collagen, type I, alpha 1 (*COL1A1*) and increased the mRNA levels of the cell cycle regulators CDK-1 and cyclin A2 (Fig 2F). In addition, p21 knockdown caused a significant increase (30-fold) in the mRNA levels of TNF-α accompanied by an increase in Smad-7 mRNA (Fig 2F), both known to negatively regulate TGF-β signaling. Overall, it appears that p21-knockdown DIS cells exhibit alterations in the cell cycle, TGF-β, DNA damage response, and focal adhesion signaling pathways.

## p21 regulates the DNA damage response in senescent cells

We then set out to understand the mechanism of cell death induction in senescent cells after p21 knockdown. We hypothesized that an increase in the level of the DNA damage response (DDR) after the knockdown accounts for this effect. We therefore assessed the phosphorylation levels of proteins involved in DNA damage response: ATM, CHK2, gamma H2A histone family member X (γH2AX), and p53. Immunoblotting revealed a substantial increase in phosphorylation of all these proteins in p21-knockdown DIS cells (Fig 3A). The increase was accompanied by an increase in the number of foci of phosphorylated ATM (p-ATM) (Fig 3B and C). To further assess the activation of the DDR in p21-knockdown DIS cells, we evaluated additional DDR markers, namely foci of the ATM target substrates p53-binding protein 1 (p53BP1) and γH2AX. The amounts of these foci were increased in the p21-knockdown DIS cells relative to control cells (Fig 3D–F). In order to assess the accumulation of DDR foci by an additional method, we stained growing and DIS cells following p21 knockdown and the control cells with γH2AX antibody and analyzed the cells in flow by ImageStreamX. The distribution of the cells according to the number of γH2AX foci and quantification

Figure 3.   p21 regulates the DNA damage response in senescent cells.
A   BJ human fibroblasts (proliferating, G; and DNA damage-induced senescent, DIS) were transduced with either siRNA targeting p21 (sip21) or control siRNA (siCtrl). Representative immunoblots for p-ATM, p-CHK2, γH2AX, p53, and p-p53(Ser 15) of at least two independent experiments are shown.
B   Immunofluorescence analysis of p-ATM in G and DIS cells transduced with sip21 or siCtrl; representative images of at least two independent experiments are shown. Arrows indicate positive staining.
C   Quantification of the total numbers of p-ATM foci per cell in (B).
D   Immunofluorescence analysis of γH2AX and p53BP1 in G and DIS BJ cells transduced with sip21 or siCtrl; representative images of at least two independent experiments are shown.
E   Quantification of total γH2AX foci per cell.
F   Quantification of total p53BP1 foci per cell.
G   ImageStreamX analysis. Distribution of numbers of γH2AX foci in G and DIS cells transduced with sip21 or siCtrl 3 days after siRNA washout. DIS cells stained with the secondary antibodies only and DAPI served as a negative control.
H   Comet assay. G and DIS BJ cells were transduced with sip21 or siCtrl. 72 h post-siRNA washout cells were tested for DNA damage using the comet assay. Olive tail moment values were calculated using ImageJ. Each specimen was evaluated in triplicate. At least 50 cells were analyzed per replica and the quantification is shown on the right panel. Representative DNA tails are shown on the left.
I   Immunofluorescence analysis of BrdU incorporation in G and DIS cells transduced with sip21 or siCtrl, 24 h post-siRNA washout; representative images of at least two independent experiments are shown.
J   Quantification of BrdU-positive G and DIS cells.
K   DAPI staining of DIS cells transduced with sip21 or siCtrl 4 days after siRNA washout.
L   SubG1 cell cycle analysis of DIS cells transduced with sip21 or siCtrl 3 days after siRNA washout; representative histograms of three independent experiment are shown.

Data information: Data were analyzed using Student's *t*-test. *$P < 0.05$. **$P < 0.005$, ***$P < 0.0005$. Data represent mean ± SEM ($n = 3$).
Source data are available online for this figure.

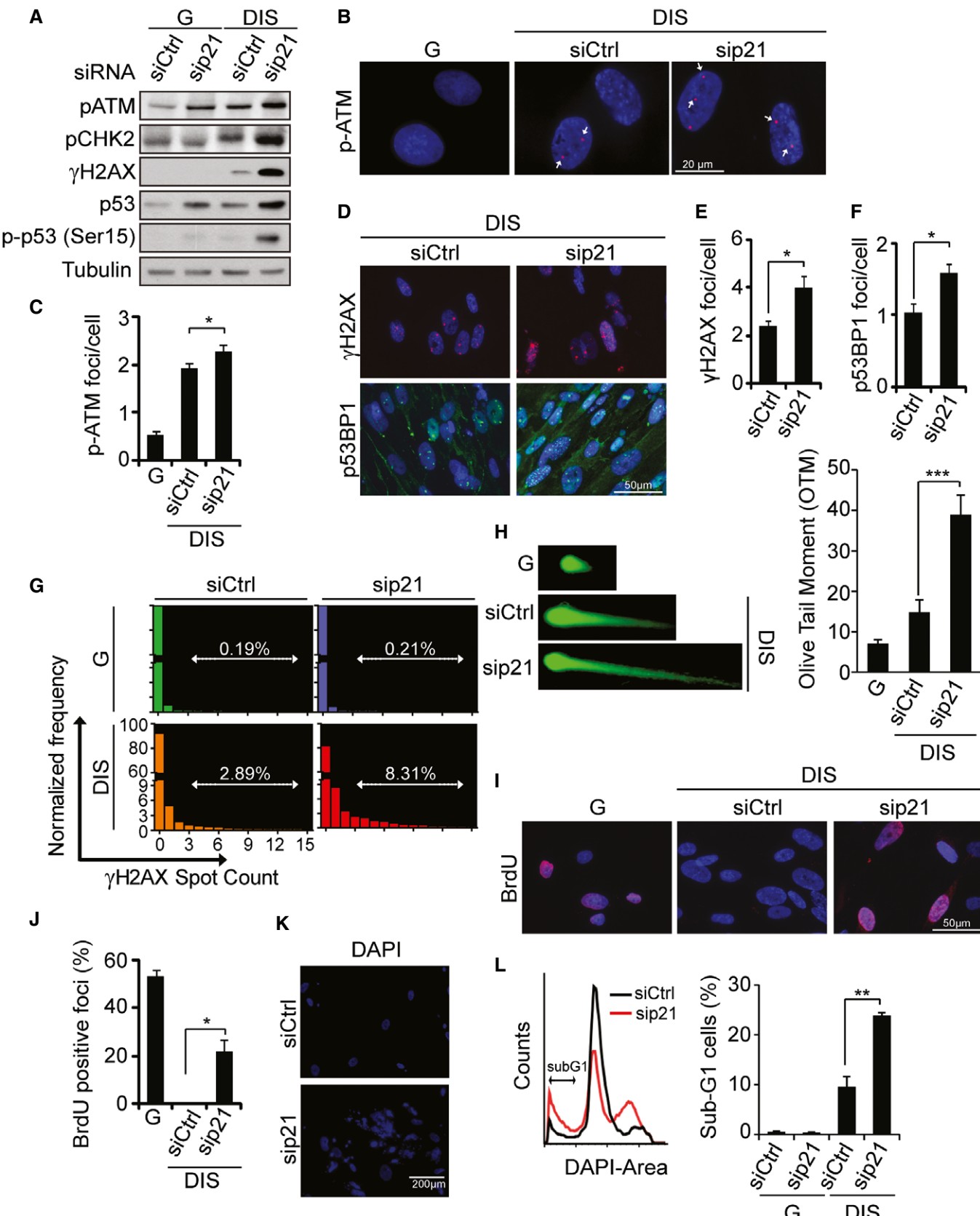

**Figure 3.**

of cells with more than two foci are presented in Fig 3G. As expected, this analysis revealed an increased in amount of DDR foci in senescent cells comparing to growing cells. p21 knockdown caused an additional significant increase in the amount of cells with more than two foci (Fig 3G, Appendix Fig S8). Next, we performed a comet assay analysis (single-cell gel electrophoresis) to measure DNA strand breaks at the level of single cells (Fig 3H). Strong increase in DNA damage following p21 silencing in DIS cells was detected, as can be seen from the representative pictures of the DNA tails and from the calculation of Olive tail moment (OTM). Therefore, p21 knockdown leads to an intensified DDR in DIS cells.

We considered the possibility that the observed induction of DDR is a result of the attempt of the p21-knockdown cells to re-enter the cell cycle. To examine this hypothesis, we assessed DNA synthesis in p21-knockdown DIS cells by using 5-bromo-2'-deoxyuridine (BrdU) incorporation. Positive BrdU staining was detected in p21-knockdown DIS cells but not in control DIS cells 24 h following the introduction of sip21 or siCtrl (Fig 3I and J). It therefore appears that ablation of p21 induces resumption of DNA synthesis. Four days after p21 knockdown, DIS cells at metaphase indeed displayed extremely fragmented chromosomes and striking micro-nucleation that was not detected in siCtrl cells (Fig 3K). The cell cycle analysis of DIS cells following p21 knockdown revealed a significant increase in the sub-G1 fraction in sip21 transduced cells comparing to siCtrl transduced cells (Fig 3L). The cell cycle plot also suggests that some of the cells in the sip21 sample appear in S-phase, in concordance with the increase in BrdU incorporation we observed in these cells. These findings might reflect entry into mitosis with numerous double-strand breaks upon cell cycle reactivation. Overall, p21 knockdown enhances the DDR in DIS cells, and this occurs at least partially via resumption of DNA synthesis.

## Cell death of DIS cells after p21 knockdown is caspase- and JNK-dependent

Accumulation of p53 establishes the growth arrest of senescence at least partially by inducing expression of p21 (Campisi & d'Adda di Fagagna, 2007). p21 in turn suppresses the phosphorylation and hence inactivates pRb, a regulator of the E2F family of transcription factors. p53 and E2F1 may therefore participate in apoptosis induction. To understand the role of these pathways in the observed phenotype, we examined whether the death of DIS cells after p21 silencing is mediated by p53 or by pRb/E2F. To test this hypothesis, we knocked down p53 or pRb individually or together with p21. Surprisingly, knockdown of p53 reduced DIS cell viability, but to a lesser extent than the reduction observed after individual knockdown of p21 (Fig 4A). This may result from the observed reduction in p21 protein following p53 knockdown (Fig 4A). The effect of combined knockdown of p21 and p53 on cell death did not differ from that of p21 knockdown alone, as detected from cell survival and cleavage of the apoptosis effectors caspase-3 and PARP (Fig 4A). It thus seems that p21 knockdown might induce death of senescent cells in the absence of p53. Similarly, the effect of combined knockdown of pRb and p21 on senescent cell viability did not differ from that of p21 knockdown alone (Fig 4B). Surprisingly, when only pRb was knocked down, it did not significantly affect cell viability, but rather caused an increase in p21, possibly suggesting a feedback loop in the regulation of the protein levels of p21. This finding suggests

that an increase in E2F activity as a result of pRb knockdown might not by itself be sufficient to induce death of senescent cells.

Extensive DNA lesions have been reported to trigger cytokine secretion and programmed cell death independently of p53 (Biton & Ashkenazi, 2011; Lanigan et al, 2011). When DNA damage is excessive, ATM drives the activation of NF-κB to accelerate TNF-α production and to mediate phosphorylation of receptor-interacting kinase 1 (RIP1). RIP1 kinase promotes the JNK-mediated induction of interleukin (IL)-8 production and the recruitment of the FADD adapter protein to activate caspase-8 and trigger programmed cell death (Biton & Ashkenazi, 2011; Condiotti et al, 2014). In view of the extensive DDR that we observed after p21 knockdown in DIS cells, we evaluated activation of the above TNF-α-driven pathway in these cells. In DIS cell, phosphorylation of JNK was substantially increased (Fig 4C). Knockdown of p21 in these cells induced an increase in phosphorylation of the p65 subunit of NF-κB (Fig 4C). We also detected pronounced expression of TNF-α in response to p21 knockdown, and this was followed by the production of IL-8 (Fig 4D). It therefore appears that the activity of NF-κB is increased after p21 knockdown in DIS cells.

We then examined whether the death of DIS cells after p21 knockdown is NF-κB dependent and is driven by the TNF-α and TNFR1 signaling loop. Combined knockdown of p21 and p65 did not rescue the cells from death (Fig 4E). It did, however, reduce cleavage of the apoptosis effectors caspase-3 and PARP (Fig 4E) and completely blocked TNF-α production (Fig 4F). Moreover, the death of DIS cells after the combined knockdown of p21 and p65 was now also accompanied by an increase in JNK phosphorylation (Fig 4E). We concluded that although NF-κB is activated after p21 knockdown in DIS cells and drives TNF-α secretion to activate the extrinsic apoptotic pathway, its silencing is not sufficient to block the death of DIS cells, probably because of the increase in JNK activity. To further characterize the cell death that follows p21 knockdown, we treated DIS p21-knockdown cells and control cells with the pan-caspase inhibitor Q-VD-OPh (QVD) or with DMSO as a vehicle control. Treatment with the inhibitor rescued cell viability only partially (17% increase in cell survival with QVD relative to DMSO; Fig 5A), whereas the activity of caspase-3 was completely blocked, as indicated by its cleavage. Similar results were obtained when caspases were blocked by the pan-caspase inhibitor z-VAD-fmk (z-VAD; Appendix Fig S9). Given that the pan-caspase inhibitors are not able to completely rescue DIS cells after p21 knockdown, we reasoned that the knockdown induces cell death via both caspase-dependent and caspase-independent mechanisms.

Activation of the JNK pathway might induce cell death that is caspase independent (Lanigan et al, 2011; Condiotti et al, 2014). We considered the possibility that the observed increase in JNK activity induces caspase-independent death of DIS cells after blockage of apoptosis with QVD. To test this hypothesis we applied the JNK inhibitor SP600125 to DIS and control cells with and without p21 knockdown. While a reduction in JNK phosphorylation level was detected, JNK inhibition by itself had no effect on the viability of DIS cells regardless of p21 knockdown (Fig 5B). The cell death that followed p21 knockdown observed in this experiment was accompanied by cleavage of the apoptosis effectors caspase-3 and PARP, indicating that it might happen through apoptosis (Fig 5B). Importantly, combined treatment with QVD and JNK inhibitor resulted in complete rescue of the senescent cells from death

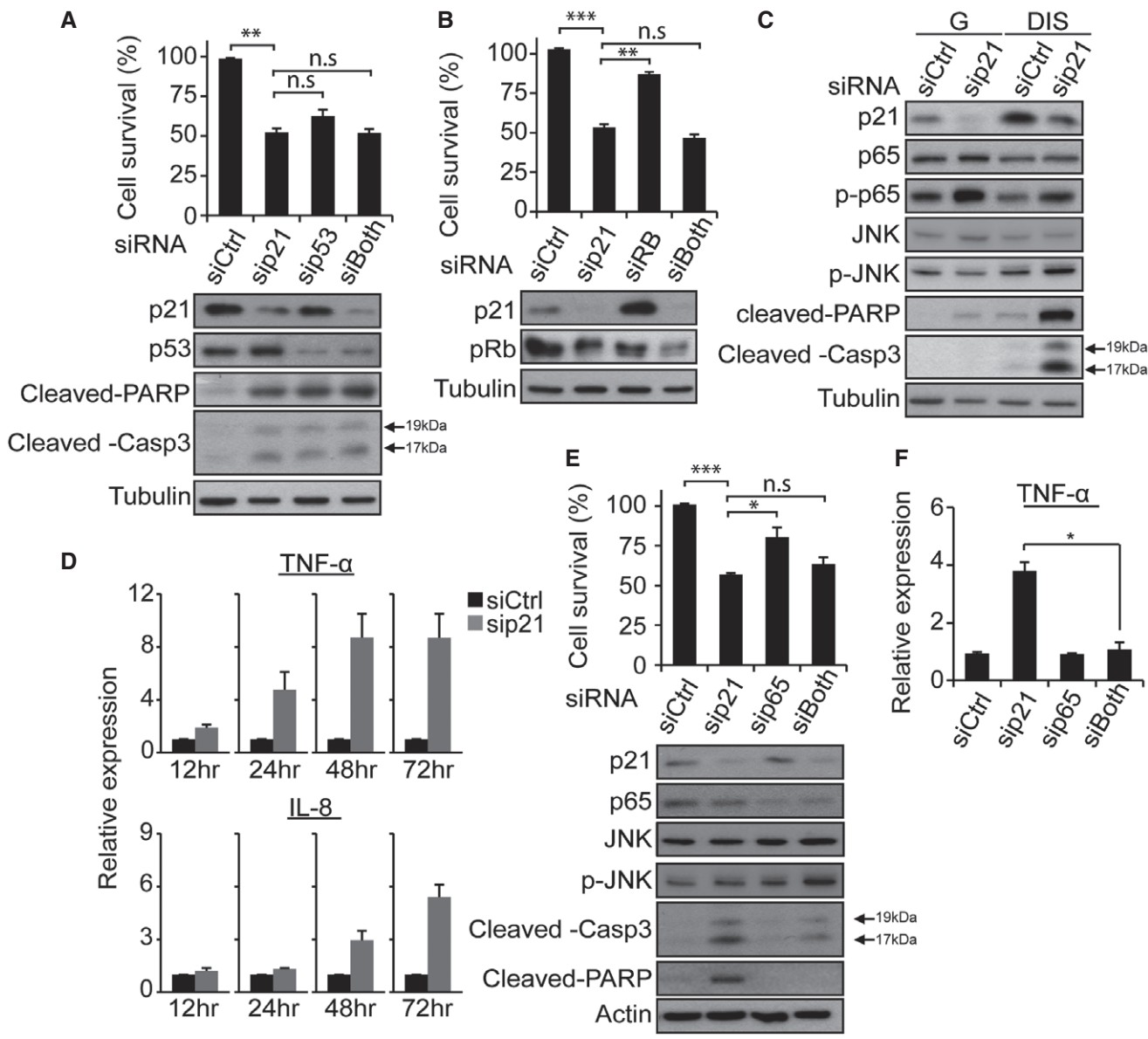

**Figure 4.  Death of DIS cells after p21 knockdown ensued by TNF-α secretion.**

A   Survival of DIS BJ cells transduced with siRNAs targeting p21, p53 or their combinations, as indicated. Data are presented as means ± SEM of 3 repeats, each performed in triplicate. Western blots of p21, p53, cleaved PARP, and cleaved caspase-3 after siRNA treatment.

B   Survival of DIS BJ cells transduced with siRNAs targeting p21, pRb or their combinations, as indicated. Data are presented as means ± SEM of 3 repeats, each performed in triplicate. Western blots depict p21 and pRb protein levels following siRNA treatment.

C   Western blot analysis of p21, p65, p-p65, JNK, p-JNK, cleaved PARP, and cleaved caspase-3 proteins 4 days after sip21 or siCtrl treatment of G and DIS cells; representative blots of at least two independent experiments are depicted.

D   mRNA expression levels of TNF-α and IL-8 in DIS cells at the indicated times after sip21 or siCtrl treatment. Data are presented as means ± SEM of three repeats.

E   Survival of DIS BJ cells transduced with siRNAs targeting p21, p65 or their combinations, as indicated, 4 days after siRNA washout. Western blots depict levels of p21, p65, JNK, p-JNK, cleaved PARP, and cleaved caspase-3 following siRNA treatment. Data are presented as means ± SEM of three repeats, each performed in triplicate.

F   mRNA expression levels of TNF-α relative to control following treatment with siRNAs targeting p21, p65, or their combinations. Data are presented as means ± SEM of three repeats.

Data information: Data were analyzed using Student's *t*-test. *$P < 0.05$, **$P < 0.005$; ***$P < 0.0005$. n.s, not significant.
Source data are available online for this figure.

following p21 knockdown (Fig 5C). To test the involvement of the JNK pathway in the regulation of cell death following p21 knockdown independent of the pharmacological inhibitor SP600125, we

knocked down JNK alone or in combination with p21 in senescent cells. Similarly to the results obtained with the JNK inhibitor, knockdown of JNK alone had no effect on the viability of senescent cells

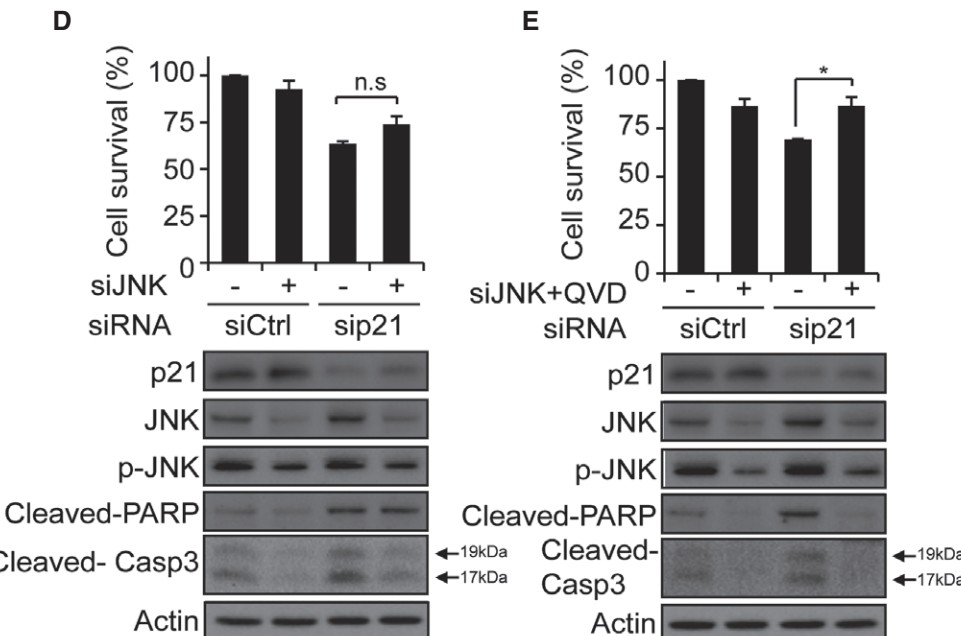

**Figure 5. Death of DIS cells after p21 knockdown is caspase- and JNK-dependent.**

A   Survival of DIS BJ cells transduced with sip21 or siCtrl, with or without incubation with pan-caspase inhibitor QVD. Western blots depict levels of p21, cleaved PARP and cleaved caspase-3 after the siRNA treatment.

B   Survival of DIS BJ cells transduced with sip21 or siCtrl, with or without incubation with the JNK inhibitor SP600125 (JNKi). Western blots depict levels of p21, JNK, p-JNK, cleaved PARP and cleaved caspase-3 after JNKi treatment.

C   Survival of DIS BJ cells transduced with sip21 or siCtrl, with or without co-incubation with QVD and JNKi. Treatment with the two inhibitors rescues the cells from sip21-induced decrease in cell viability.

D   Survival of DIS BJ cells transduced with siRNAs targeting p21, JNK1 or their combinations, as indicated. Western blots depict levels of p21, JNK, p-JNK, cleaved PARP and cleaved caspase-3 following siRNA treatment.

E   Survival of DIS BJ cells transduced with sip21 or siCtrl, with or without transduction of siJNK1 and incubation QVD. Western blots depict levels of p21, JNK, p-JNK, cleaved PARP and cleaved caspase-3 after treatments.

Data information: For all the survival plots, data are presented as means ± SEM of 3 repeats, each performed in triplicate. Data were analyzed using Student's *t*-test. *$P < 0.05$. **$P < 0.005$. n.s, not significant.

Source data are available online for this figure.

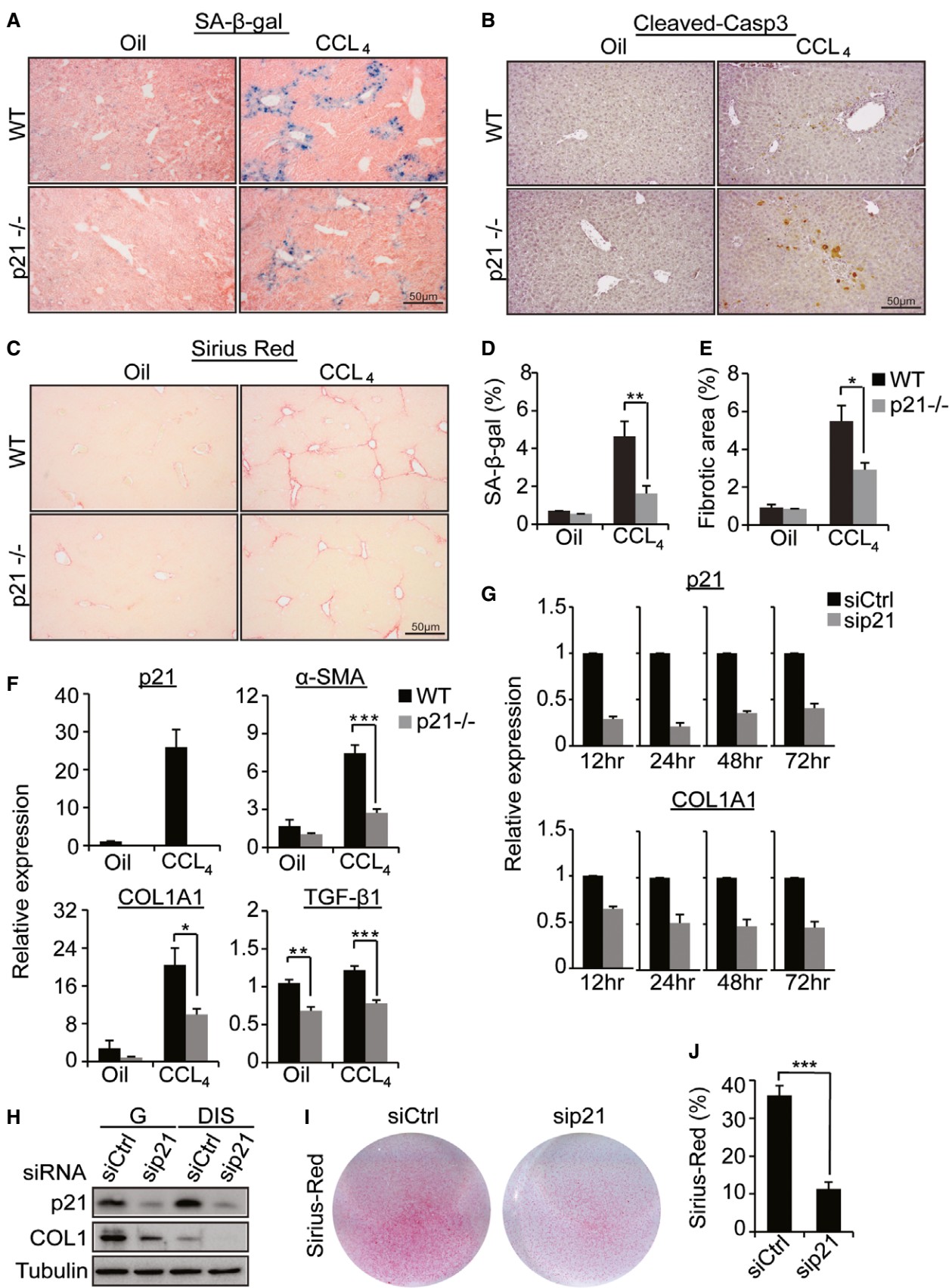

Figure 6.

(Fig 5D). It also was not able to rescue the death of senescent cells induced by the knockdown of p21 (Fig 5D). However, when JNK knockdown was performed in combination with QVD treatment, it rescued the death of senescent cells induced by the knockdown of p21 and substantially reduced the levels of cleaved PARP and caspase-3 in the p21-knockdown senescent cells (Fig 5E). Overall, it appears that p21 knockdown in DIS cells induces cell death in a caspase- and JNK-dependent manner.

### p21 knockout alleviates liver fibrosis

Knockdown of p21 causes death of DNA damage-induced senescent cells in tissue culture. We therefore aimed to determine the effect of p21 on senescent cell viability *in vivo* in a pathological condition where senescent cells, with activation of DDR, are present. One such condition is liver fibrosis, where activated HSCs become senescent to limit their proliferation resulting from liver damage (Krizhanovsky *et al*, 2008). To examine the role of p21 in this process, we subjected WT and $p21^{-/-}$ 8-week-old female mice to 6 weeks of treatment with the fibrogenic agent $CCl_4$. On the day after the last $CCl_4$ injection, livers were analyzed by SA-β-gal assay for the presence of senescent cells (Fig 6A), cleaved caspase-3 immunostaining for presence of apoptotic cells (Fig 6B) and by Sirius Red staining for fibrosis (Fig 6C). As expected, the WT mice developed liver fibrosis and an accumulation of senescent cells around fibrotic scar areas (Fig 6A). SA-β-gal staining revealed a significant decrease in the amount of senescent cells in $p21^{-/-}$ mice comparing to the WT (Fig 6A and D). Strikingly, presence of apoptotic cells was detected in the livers of $p21^{-/-}$ mice (Fig 6B) and Sirius Red staining was significantly reduced in these mice comparing to WT mice (Fig 6C and E). This reduction was accompanied by a decrease in expression of molecular markers of fibrosis in the livers of $p21^{-/-}$ mice (Fig 6F), where significant reductions were observed in the expression levels of TGF-β, COL1A1, and the smooth muscle actin α-SMA (a marker of activated HSCs) (Fig 6F). Apparently, therefore, p21 knockout alleviates liver fibrosis.

The significant effect of p21 on liver fibrosis cannot be fully explained, however, by the reduction in amount of senescent cells. We therefore evaluated the direct effect of p21 on collagen production. To this end, DIS cells were transduced with siRNAs against p21 and subjected to RT–PCR analysis of COL1A1 at 12, 24, 48, and 72 h after siRNA washout. Surprisingly, a 48% decrease in COL1A1 mRNA was already detectable 24 h after p21 knockdown, before any cell death had occurred (Figs 6G and 1E). We measured the levels of COL1 protein in DIS and control cells after p21 knockdown to determine whether the reduction in COL1 expression is limited to DIS cells. A reduction in COL1 protein was detected in both growing and DIS cells (Fig 6H). The total amount of secreted ECM was then tested by Sirius Red staining after p21 knockdown in cultured BJ cells. The p21 knockdown led to a significant decrease in ECM secretion (Fig 6I and J). Overall, it seems that p21 knockout alleviates liver fibrosis by a mechanism that may involve a decrease in the number of senescent activated HSCs as well as the direct down-regulation of collagen.

## Discussion

Cellular stress induced by persistent DDR is a central mechanism that drives senescence in aging and age-related diseases (Campisi, 2013; Munoz-Espin & Serrano, 2014; Ovadya & Krizhanovsky, 2014). DDR induces p21 expression via p53. Upregulation of p21 is a molecular mechanism that enables DIS cells to maintain their viability after damage induction, and allows their retention within tissues. Silencing of p21 enhances DDR in DIS cells via resumption of DNA synthesis. The enhancement of DDR is accompanied by an increase in activation of ATM, which drives NF-κB to activate JNK via the TNF-α/TNFR1 signaling loop (Appendix Fig S10). Thus, the DDR-mediated death of DIS cells after p21 knockdown is dependent on activation of caspase and JNK.

Following p21 knockdown in DIS cells, more than one mode of cell death involving both caspases and JNK might be activated. Both caspases and JNK can be activated by DNA damage in parallel to the NF-kB activation (Biton & Ashkenazi, 2011). Moreover, p21 might directly regulate activity of JNK, upstream kinase ASK1 and procaspases (Abbas & Dutta, 2009). Therefore, it is possible that while caspase cleavage is dependent on NF-kB activity, JNK activity is independent of this pathway. Indeed, the knockdown of p65 alone does not affect JNK phosphorylation and does not rescue DIS cell death induced by p21 knockdown. We cannot exclude the involvement of direct regulation of JNK phosphorylation and caspases by p21 in our system. However, more complex mechanism might be based on the observation that JNK activation shifts the balance of TNF-stimulated cell death from apoptosis to necrosis (Ventura *et al*, 2004). In addition, when apoptosis is blocked, various apoptotic stimuli can activate autophagy and JNK to induce autophagic cell death (Choudhury *et al*, 2007; Bedelbaeva *et al*, 2010; Serrano, 2015). Senescent cells, however, are resistant to apoptosis (Wang,

---

◄  **Figure 6.  p21 knockout alleviates liver fibrosis.**

A–C    Liver sections from wild-type (WT) or p21-knockout mice ($p21^{-/-}$) treated with $CCl_4$ (*n* = 10 from each group) or vehicle (oil; *n* = 3 (WT) and *n* = 4 ($p21^{-/-}$)) were stained with SA-β-gal (A) cleaved caspase-3 (B) and Sirius Red (C).

D       Quantification of SA-β-gal staining of sections presented in (A).

E       Quantification of Sirius Red staining presented in (C).

F       Relative expression levels of mRNA of p21, the HSC marker α-SMA (ACTA2), and the fibrosis markers COL1A1 (collagen, type I, alpha 1) and TGF-β1 in WT and $p21^{-/-}$ fibrotic mouse livers.

G       Relative expression levels of mRNA in p21 and COL1A1 in BJ DIS cells after transduction with sip21 and siCtrl at the indicated time points.

H       Western blot analysis of p21 and collagen-1 (COL1) proteins after sip21 and siCtrl treatment of G and DIS cells.

I       Sirius Red staining of DIS cells after transduction with sip21 and siCtrl.

J       Quantification of Sirius Red staining presented in (H).

Data information: Data were analyzed using Student's *t*-test. *$P$ < 0.05. **$P$ < 0.005. ***$P$ < 0.0005. Data represent mean ± SEM (*n* = 3).
Source data are available online for this figure.

1995; Sanders *et al*, 2013; Yosef *et al*, 2016). These cells use autophagy for their maintenance (Young *et al*, 2009) and show a significant increase in TNF-α production after p21 knockdown. Therefore, we speculate that treatment with the pan-caspase inhibitor QVD, while blocking apoptosis, might allow necrotic or autophagic cell death in DIS cells following p21 knockdown, thereby preventing the rescue of the cells from death. Further research will be required in order to define the mode of cell death induced in senescent cells after knockdown of p21.

Persistent DDR therefore supports senescent cell viability in a cell-autonomous manner by inducing the expression of p21. However, DDR also participates in controlling regulation of the expression of immune ligands on senescent cells that mediate their recognition and consequent elimination by NK cells (Sagiv *et al*, 2016). The interaction of senescent cells with NK cells is regulated by complex mechanisms involving intercellular protein transfer (Biran *et al*, 2015), which can also be affected by DDR. Therefore, persistent DDR might be a master regulator that limits cell-autonomous cell death of senescent cells, but drives their recognition and elimination by the immune system. Interestingly, OIS cells while also characterized by activation of DDR (Di Micco *et al*, 2006), were insensitive to the p21 knockdown. This effect might be mediated by direct pro-survival effect of activated Hras (Bonni *et al*, 1999) or by strong increase in the expression of p16 in OIS cells which might be able to substitute for p21 or inhibit JNK signaling directly (Choi *et al*, 2005).

In liver fibrosis activated stellate cells become senescent and exhibit activation of DDR. Elimination of these senescent cells by NK cells is needed to restrain liver fibrosis and return the liver to a predamaged state following short-term damage (Krizhanovsky *et al*, 2008). We showed here that presence of senescent cells in the fibrotic liver was diminished in p21$^{-/-}$ mice. This finding represents a first demonstration of efficient reduction in the number of senescent cells in a living tissue by gene knockout. This reduction might be a result of increased death of senescent cells, or alternatively, of their decreased formation owing to a shift from senescence to apoptosis in p21$^{-/-}$ cells following their exposure to senescence-inducing stimuli. The reduction in presence of senescent cells in fibrotic livers was accompanied by an alleviation of the fibrotic process. Fibrosis is regulated by an intricate interplay between TGF-β pro-fibrotic and TNF-α anti-fibrotic signaling pathways (Henderson & Iredale, 2007; Puche *et al*, 2013). Owing that non-senescent cells express lower levels of COL1 following p21 knockdown, reduction in fibrosis in p21$^{-/-}$ mice might be a result of a cell-autonomous inhibitory effect of p21 knockout on TGF-β signaling and collagen production in HSCs. Alternatively, increased TNF-α secretion caused by p21 knockout in senescent cells can affect activation of HSCs in a cell non-autonomous manner via the ability of TNF-α to antagonize the pro-fibrotic effect of TGF-β (Olive *et al*, 2008). Alternatively, p21 deletion might be able to improve hepatocyte proliferation in the context of CCl$_4$ induced DNA damage. In that case, apoptosis may affect some hepatocytes continuing to proliferate in the presence of DNA damage. It is also possible that the effect of p21 knockout on liver fibrosis derives from its ability to influence the viability of senescent cells and to regulate the microenvironment by directly controlling the production of ECM components and by secreting TNF-α. Interestingly, levels of pJNK are increased in activated stellate cells during liver fibrosis, similarly to DIS, and treatment with

JNK inhibitor alleviated liver fibrosis in the mouse model we used in our study (Kluwe *et al*, 2010). Notably, TGF-β and TNF-α regulate fibrotic processes in all tissues in the organism. Therefore, p21 inhibition might be an important strategy for alleviating fibrotic processes and fostering tissue regeneration, as a result of elimination of senescent cells.

Senescent cell elimination can indeed increase the proliferative capacity of stem cells and hence support tissue regeneration (Chang *et al*, 2016; Yosef *et al*, 2016). Elimination of senescent cells may therefore allow better recovery of tissues, consistent with the finding that senescent cell elimination via transgenic techniques improves aging-associated phenotypes (Baker *et al*, 2011, 2016). Interestingly, p21 knockout was previously shown to support tissue regeneration through various mechanisms. As an example, knockout of p21 prolongs the lifespan of telomerase-deficient mice by rescuing the proliferation of intestinal progenitor cells and improving the repopulation capacity and self-renewal of hematopoietic stem cells (Choudhury *et al*, 2007). In addition, p21 deficiency supports tissue regeneration, including complete rescue of regeneration of the impaired liver (Stepniak *et al*, 2006; Bedelbaeva *et al*, 2010; Lehmann *et al*, 2012). p21 also plays a complex role in senescence and aging in BubR1 hypomorphic progeroid mice (Baker *et al*, 2013). It thus seems that p21 might be a master regulator of tissue regeneration through its control over a combination of cellular and molecular pathways. Overall, elimination of senescent cells by inhibition of p21 appears to be a promising strategy for targeting senescent cells during and after tissue damage and in age-related diseases. Such targeting will employ all combinations of pathways regulated by p21 to promote tissue fitness.

# Materials and Methods

### Cell culture

Human IMR-90 and BJ fibroblasts were obtained from the American Type Culture Collection. Mouse embryonic fibroblasts (MEFs) were isolated according to standard procedures (McCurrach & Lowe, 2001). Cells were maintained in DMEM supplemented with 2 mM L-glutamine, 100 units/ml of penicillin, 100 μg/ml of streptomycin, and 10% fetal bovine serum (FBS). DNA damage-induced senescence (DIS) was introduced by treatment with etoposide (E1383, Sigma-Aldrich) at a concentration of 20 μM (for MEFs and H1299) or 50 μM (BJ and IMR-90) for 48 h (please see Appendix Fig S1 for the timeline of the treatment). Oncogene-induced senescence (OIS) was introduced by means of retroviral infection with mutated HRas$^{v12}$ (pLPC mCherry-H-Ras 12V plasmid 45) at 3% oxygen. Empty vector-transduced cells served as controls. Retroviruses were packaged, and infections were performed as described (Narita *et al*, 2006). The infected population was selected using 2 μg/ml puromycin (Sigma) for 2 days. By 7 days post-selection, the cells had acquired the senescence phenotype.

### Cell survival assays following siRNA treatments

Target cells were plated in 12-well plates. Cells were transfected with ON-TARGETplus SMARTpool small interfering RNA (siRNA)

targeting p21 (L-002000-00-0005), JNK1 (L-003514-00-0005) or, as a control, with non-targeting siRNA pool (D-001810-10-20), using DharmaFECT formulation 1 reagent (T-2001-03) according to the instructions of the manufacturer (Thermo Fisher Scientific). At 4 days post-transfection, or at the indicated time point, the percentages of remaining adherent cells were quantified relative to control-treated cells, using PrestoBlue reagent (A13262, Life Technologies). When indicated, the following molecules were added to each well following siRNA washout: z-VAD-FMK (A1902, ApexBio) was added to the cells at a concentration of 100 μM with daily replenishment; Q-VD-Oph (A1901, ApexBio) was added to the cells 24 h after siRNA washout for an additional 48 h at a concentration of 50 μM; JNK inhibitor SP600125 (BML-EI305, Enzo Life Sciences) was added to the cells 24 h after siRNA washout for an additional 48 h at a concentration of 5 μM.

### Cell survival following shRNA transduction

H1299 cells were transduced with vector encoding validated GFP-linked small hairpin RNA (shRNA) targeting p21 (shp21; Chicas et al, 2010) or with shRNA specific for luciferase gene (shLuc) as control. Following selection with puromycin, cells were treated with etoposide. Cell viability was determined 7 days post-washout using PrestoBlue reagent.

### Cell cycle analysis

Cultured cells were collected together with their supernatant, and fixed with 80% ice-cold ethanol. Following overnight incubation in −20°C, cells were washed twice with cold PBS, incubated for 10 min in 0.1% Triton solution (in PBS), followed by staining with DAPI (1 μg/ml). Samples were filtered to remove aggregates prior to analysis. DAPI area and width were measured by flow cytometry (BD Biosciences FACSCalibur). Images were analyzed using FlowJo software (Treestar, Ashland, OR).

### ImageStreamX analysis

Cells were collected and fixed with fixation buffer (eBioscience) for 30 min at 4°C. Then, cells were washed with permeabilization buffer (eBioscience), and incubated overnight with phospho-histone H2A.X antibody (Ser139; Cell Signaling #2577). Cells were washed twice and incubated with secondary antibodies (Jackson Immuno Research), followed by two additional washings and staining with DAPI for 10 min. Cells were imaged by ImageStreamX mark II (Amnis-EMD Millipore) and γH2AX foci were quantified using IDEAS 6.1 software (Amnis). Normalized frequency was defined as the frequency cells with the indicated amount of γH2AX spots out of the total number of cells.

### Comet assay

The assay was carried out according to the manufacturer instructions (Enzo, #ADI-900-166) for alkaline unwinding and electrophoresis conditions. Gel electrophoresis was carried out at 300 mA for 10 min at 4°C. Positive control cells were treated with 100 μM hydrogen peroxide for 20 min. Positive and negative controls were run along with each sample. A minimum of 50 individual cells per sample were scored in duplicates. Comet analysis was performed using the ImageJ analysis tool.

### Sirius red staining for tissue culture

Cell layers were fixed with 4% paraformaldehyde for 20 min at room temperature. The solution was removed and plates were washed in running tap water. The plates were then air-dried, and Sirius Red dye solution (1 mg/ml in picric acid) was added to each well for 1 h with mild shaking. Each well was washed four times with 2 ml of 0.01 N HCl to remove unbound dye.

### Immunoblotting

Cell lysates (15–30 μg of protein) were resolved by 15% SDS–PAGE and transferred onto Immobilon P membranes (IPVH00010, Millipore). After blocking of the membranes with 5% bovine serum albumin (BSA) in TBST (Tris-buffered saline with 0.01% Tween-20) for 1 h, they were probed with antibodies against cleaved PARP (#9541), cleaved caspase-3 (#9661), JNK (#3708), phospho-Chk2 (Thr68; #2661), phospho-histone H2A.X (Ser139; #2577), phospho-JNK (Thr183/Tyr185; #4668), phospho-p53 (Ser15; #9284), phospho-p65 (Ser536; #3033), Rb (#9313) (all from Cell Signaling Technology), p65 (sc-372) and β-tubulin (sc-9104) (both from Santa Cruz Biotechnology), p21 (556431, BD Biosciences), collagen-1 (600-401-103-0.5, Rockland Immunochemicals), phospho-ATM (Ser1981; 2152-1, Epitomics), p16 (ab108349, Abcam) p53 (mix of DO-1 and PAb1801, kindly provided by M. Oren, Weizmann Institute of Science) and β-actin (A5441, Sigma-Aldrich). Antibodies were visualized by chemiluminescence detection (34080, Thermo Fisher Scientific).

### Microarray analysis

Total RNA was extracted from growing and DIS BJ cells transfected with siRNA against p21, or with control siRNA, using an RNeasy Mini Kit (QIAGEN). RNA purity was assessed with an ND-1000 NanoDrop Spectrophotometer (Peqlab Biotechnologie) and BioAnalyzer 2100 system (Agilent Technologies). cDNA was prepared, labeled, and hybridized to Affymetrix GeneChip PrimeView, Human Gene Expression Array, according to standard manufacturer protocols. Hybridized chips were stained, washed, and scanned with the Affymetrix GeneChip 3000 7G Plus scanner and converted to cell intensity files by Affymetrix Expression Console Software™. Transcriptome analysis (processing of raw data, clustering, and creation of correlation matrices) was carried out with Partek Genomics Suite 6.6 (http://www.partek.com/). The raw probe intensities were adjusted according to the numbers of G and C bases in the probe sequence before any probe correction. Preprocessing was performed with the Robust Microarray Averaging algorithm (Irizarry et al, 2003). Genes expressed below background levels (log 1.5 intensity of 6.9) in all examined conditions were excluded from further analyses. To perform functional enrichment tests of the candidate genes we used WebGestalt (Falcone et al, 2013) for WikiPathways analysis and the Ingenuity Pathway Analysis (IPA) system (Merino et al, 2012) for both canonical pathways and molecular networks.

## Quantitative RT–PCR

Total RNA was extracted using an RNeasy kit (74104, QIAGEN), followed by DNase I treatment. cDNA was produced using random hexamers. The cDNA samples were amplified using Platinum SYBR Green qPCR SuperMix (11744-500, Life Technologies) in a StepOnePlus Real-Time PCR System (Applied Biosystems). Relative expression was normalized using the expression levels of GAPDH or actin.

## Bromodeoxyuridine (BrdU) incorporation and immunofluorescence

When indicated, growing and DIS BJ cells were pulsed with 0.1 mg/ml 5-bromo-2′-deoxyuridine (BrdU; B-9285, Sigma-Aldrich) and 0.1 mg/ml 5-fluoro-2′-deoxyuridine (FDU, F-0503, Sigma-Aldrich) for 24 h. The cells were fixed and processed for immunofluorescence with monoclonal anti-BrdU antibody (555627, BD Biosciences), phospho-H2AX (#2577, Cell Signaling), phospho-ATM (sc-47739, Santa Cruz), and 53BP1 (NB100-304, Novus Biologicals).

## Mice

p21$^{-/-}$ mice (Cdkn1a<tm1Led>/J) were obtained from the Jackson Laboratory (#016565) and maintained on C57Bl/6 background. Wild-type C57Bl/6 (WT) and p21$^{-/-}$ 8-week-old female mice were treated with 12 consecutive biweekly i.p. injections of 1 ml/kg CCl$_4$ to induce liver fibrosis, as described (Krizhanovsky *et al*, 2008). Ten mice of each genotype are sufficient to identify statistical differences according to the previous studies. The mice were assigned to the experimental groups according to their genotype without randomization or blinding. Animals were killed 24 h after the last injection, and their livers were embedded in paraffin for immunohistology, frozen in OCT solution for cryosectioning and senescence-associated beta-gal (SA-β-gal) staining, or homogenized for RNA and protein extraction. All procedures described in this work were approved by The Weizmann Institute of Science Animal Care and Use Committee.

## Histological analysis

Immunohistology was performed on 5-μm paraffin sections according to standard procedures. Sections were deparaffinized and rehydrated in an ethanol series. Antigen retrieval was performed in a boiling water bath, and sections were blocked for non-specific binding with 4% horse serum and 1% BSA. Primary antibodies recognizing cleaved caspase-3 (#9661; Cell Signaling) were applied overnight at 4°C. Staining was developed using DAB (Vector Laboratories) followed by hematoxylin counterstaining. Paraffin-embedded tissue sections were stained with hematoxylin–eosin for routine examination or with Sirius Red for visualization of fibrotic deposition. These images were quantified using NIH ImageJ software (http://rsb.info.nih.gov/ij/). We calculated the amount of fibrotic tissue in diseased mice relative to the basal amount of Sirius Red staining present in normal mouse liver. For SA-β-gal staining, 10–12-μm cryosections of OCT-embedded mouse livers were fixed in 0.5% glutaraldehyde for 15 min and stained as previously described (Krizhanovsky *et al*,

2008). Sections were visualized using an Olympus microscope, and images were analyzed using CellP software (Diagnostic Instruments).

## Statistical analysis

Data are presented as means ± SEM. Statistical significance was determined using Student's *t*-test. A *P*-value of < 0.05 was considered significant.

**Expanded View** for this article is available online.

## Acknowledgements

We thank I. Ben-Porath for insightful suggestions; I. Orr for helping analyze the microarray data; M. Oren for anti-p53 antibodies; and all the members of Krizhanovsky laboratory for stimulating discussions. This work was supported by grants to V.K. from the European Research Council under the European Union's FP7, Israel Science Foundation, and Bruce Kanter Fund. V.K. is the incumbent of the Carl and Frances Korn Career Development Chair in Life Sciences.

## Author contributions

RY, NPi, NPa, HG, and EV designed and performed experiments. YO and SM performed experiments. SB-D analyzed the microarray data. ZP analyzed ImageStream experiments. RY and VK analyzed the experiments and wrote the manuscript. VK supervised the project.

## Conflict of interest

The authors declare that they have no conflict of interest.

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
