## [Review Process File · The EMBO Journal]

Manuscript EMBO-2016-95553

p21 maintains senescent cell viability under persistent DNA damage response by restraining JNK and caspase signaling

Reut Yosef, Noam Pilpel, Nurit Papismadov, Hilah Gal, Yossi Ovadya, Ezra Vadai, Stav Miller, Ziv Porat, Shifra Ben-Dor and Valery Krizhanovsky

Corresponding author: Valery Krizhanovsky, Weizmann Institute of Science

Review timeline:

Submission date:	21 August 2016
Editorial Decision:	07 October 2016
Revision received:	21 February 2017
Editorial Decision:	22 March 2017
Revision received:	04 May 2017
Accepted:	08 May 2017

Editor: Daniel Klimmeck

Transaction Report:

1st Editorial Decision

07 October 2016

Thank you for the submission of your manuscript (EMBOJ-2016-95553) to The EMBO Journal. Your study has been sent to three referees, and we have received reports from all of them, which I copy below.

As you will see, the referees acknowledge the potential high interest and novelty of your work, although they also express a number of major concerns that will have to be addressed before they can support publication of your manuscript in The EMBO Journal. In particular, referee #2 states that the current evidence for an achieved senescent state after etoposide treatment is not convincing in his/her view and asks you to provide further molecular proof (ref #2, specific comment). Referee #1 agrees that an extended characterization of senescence would be needed to corroborate the findings (ref #1, pt. 1). In addition, both referees #1 and #3 state that further results are not sufficiently well supported by the data or lack conclusiveness and thus should to be revised (ref #1, pts. 3,4; ref #3 pt. 1). In addition, all referees list a number of technical issues and controls that would need to be addressed to achieve the level of robustness needed for The EMBO Journal.

I judge the comments of the referees to be generally reasonable and we are in principle happy to invite you to revise your manuscript experimentally to address the referees' comments. I agree that it would be essential to consolidate the senescence phenotype. Please note that a revised version of the manuscript would be sent back to all three referees and acceptance of the manuscript will essentially depend on the completeness of your responses included in the next version of the manuscript.

Please feel free to contact me if you have any questions or need further input on the referee

comments.

Thank you for the opportunity to consider your work for publication. I look forward to your revision.

 REFEREE REPORTS

Referee #1:

In this article, Yosef et al. demonstrated that CDK inhibitor p21 maintains viability of DNA damage-induced senescent (DIS) cells and p21 knockdown (or knockout) leads to their elimination. Authors compared gene expression of DIS cells with and without p21 knockdown and investigated different affected pathways, focusing on DNA damage response. They concluded that the decrease of DIS cell survival is p53-independent and the result of ATM-NFκB-TNFα pathway activation, which leads to death of senescent cells in JNK- and caspase-dependent manner. Moreover, authors demonstrated reduced amount of senescent cells in p21^{-/-} mouse livers subjected to fibrogenic agent CCl₄. Importantly, the fibrotic scars were diminished as well, adding one more evidence of non-beneficial functions of p21 and further enhancing the concept of advantages of anti-p21 treatment. The study touches on an area of general interest. The authors present some very interesting culture based work on a role of p21 in promoting survival of senescent cells by inhibiting cell cycle progression in the presence of DNA damage and thus by inhibiting a NFκB dependent TNFα induction. The *in vivo* work is also interesting but has some major flaws. It is well possible that p21 deletion improves the proliferative capacity of hepatocytes in mice exposed to repeated doses of CCl₄. Activation of stellate cells in the context of chronic liver injury is triggered by the regenerative decline of hepatocytes. Therefore, it is well conceivable that improved hepatocyte proliferation in response to p21 deletion prevents the induction of stellate cell activation and the induction of stellate cell senescence rather than promoting the death of senescent stellate cells.

Specific points:

1) Fig. 1. Just mentioning that "the cells had acquired the senescence phenotype" is not enough. SA-β-gal staining, proliferation curves and FACS plots of cell cycle distribution before and after etoposide washout, as well as other markers of senescence should be sufficiently provided in supplementary figure.

Minor comment: It is surprising that authors did not see the difference between control samples in graphs of Fig. 1A,B,C. Since PrestoBlue reagent, like any other similar reagents, estimates the amount of metabolically active cells, proliferating G and V groups must show higher values than senescent DIS and OIS. It seems that authors normalized to 100% all control samples, which is mathematically wrong to do since 100% in G group is not the same 100% in DIS group because of different normalization coefficient. So, either all values have to be normalized with one coefficient (like in Fig. 4G,H,I) or all groups have to be presented in separate plots, so nobody will compare controls between each other.

2) Fig. S1. Experiment with p53-null H1299 cell line is not described in Materials & Methods. Transduction protocol, etoposide dose, period of treatment and acquiring time of senescent phenotype have to be explained. Similar to comment 1, markers of senescence have to be provided.

3) Fig. 3. From my point of view, this part of the study is rather not convincing and should be strengthened. More experiments should be performed to demonstrate DNA lesions and DNA-damage response after sip21. Comet assays, FACS analysis of sub-G1 percentage, WBs of involved proteins and FACS for quantitative estimation of γH2AX and 53BP1 could be done.

a) Fig. 3A. Opposite to Fig. 4A, p53 protein in Fig. 3A is clearly stabilized after sip21 and demonstrate dramatic Ser-15 phosphorylation, which should be at least discussed, since authors claim that the decreased survival of DIS cells after sip21 is p53-independent. p53 phosphorylation is indeed dramatic, although amount of p-ATM foci increased from 2 to just 2.2 foci/cell, probably meaning another involved mechanism.

b) Fig. 3I. Since in this figure DAPI staining demonstrated such dramatic discrepancy with any other DAPI pictures, differences in sample preparations (time after sip21, etc.) should be explained in the figure legend. As mentioned above, additional methods for DNA lesion detection should be performed.

Minor comment: Typo in Fig. 3F - "53BP1", not "p53BP1".

4) Fig. 4 raises a number of questions:

- a) Fig. 4A. Authors claim "that p21 maintains the viability of senescent cells downstream of p53" and in p53-independent manner. This conclusion can hardly be obtained from provided data, since sip53 does not reduce p21 amount in comparable with sip21 level but the survival of DIS cells after sip53 is similar to p21 knockdown. How authors explain such dramatic decrease of cell survival after sip53? Other targets of p53?
- b) Fig. 4B,E. There is a number of confusing and non-matching WB results. Why do pRb amount decreased after sip21 and, oppositely, the level of p21 dramatically increased after sipRb? Should be explained. Why there is a strong band of Cleaved-PARP in siCtrl sample in Fig. 4E, contradicting all other results in Fig.4E (is it the same experiment?)?
- c) Fig. 4C,E. It is clear that p-p65 is increased after sip21 even without DIS. How authors would explain the difference in JNK and caspase activations in G and DIS groups? As well mechanism of increased JNK activation after p65 knockdown (Fig. 4E) should be explained. Importantly, total JNK level has to be provided as a control for phosphorylation blots (same for Fig. 4H).
- d) Fig. 4D & 5F. Similar to comment 1, all values have to be normalized with only one coefficient or all groups have to be presented in separate histograms.
- e) Fig. 4H,I. It is known that chemical inhibitors in general and MAPK inhibitors in particular usually demonstrate unspecific action (particularly for SP600125, see Tanemura et al., *Curr. Enzym. Inhib.* 2010; 6:26-33). That is why I would recommend to prove Fig. 4H,I experiments with anti-JNK siRNAs, importantly, since involvement of JNK pathway seems to be the key result of the study.
- f) Fig. 4I. One of the most important results of the paper misses WBs. Should be provided. Minor comment: The upper huge band in cleaved caspase 3 blots in Fig. 4G and Fig. S7 should be designated.
- 5) Fig. 5. The main question here - does the decreased amount of senescent cells in p21^{-/-} livers show their excessive elimination (as stated in the abstract and introduction) or reduced formation? Principally, experiment with in vivo injections of sip21 could answer this question. Other potential players, JNK for example, could be validated as well. Minor comment: a) Mouse strain has to be designated in Materials & Methods. b) The mechanism of collagen expression regulation by p21 should be discussed.

General minor comments:

- 1) It is well known, and described in detail for example in the mentioned review of Abbas & Dutta, 2009, that p21 interacts with and thus directly regulates activity of JNK, upstream kinase ASK1 and procaspases. I would suggest that this level of regulation and its possible role in the observed effects should be considered and discussed.
- 2) There are some non-correct citations. For example, Lanigan et al., 2011 at the page 11 (3rd line from the bottom) should be replaced to Biton & Ashkenazi, 2011, and Condiotti et al., 2014 at the page 18 (5th line) should be replaced to Choudhury et al., 2007. Authors should double-check all references.
- 3) In order to make the conclusions more comprehensible, authors could consider a diagram preparation, which would sum up their model of newly defined p21 actions.
- 4) Materials & Methods. Transduction protocols should be described with more details or references should be provided. The same is true for the transfection procedure; at least transfection reagent should be mentioned. PD number of primary cells has to be stated. Phosphorylation sites for all relevant antibodies have to be named.
- 5) Statistical analysis. Instead of SEM, which just represents the mathematical preciseness of mean value calculation, standard deviation SD generally more preferable in biology, since represent data variability, which is more informative. Authors could for example look at Tom L. Croat. *Med. J.* 2004;45:361-70.
- 6) Immunoblotting analysis. Generally, there is a good practice of providing uncropped and unprocessed immunoblot pictures in supplementary figure. I would strongly recommend to do it.

Referee #2:

Yoseph et al. address a still incompletely understood question in the senescence field: what maintains senescent cell viability during the prolonged proliferative arrest? The authors demonstrate that suppressing p21 levels, either by siRNA mediated knockdown or by using p21^{-/-} MEFs/mice, results in reduced cell viability following 48h treatment with the topoisomerase inhibitor etoposide. Following drug treatment, cells with reduced p21 protein levels displayed increased levels of DNA

damage and showed a more robust activation of a DNA damage response, compared to control treated cells. Due to the inactive DNA damage response, cells with reduced p21 levels continued to proliferate, acquired genomic aberrations visible by DAPI staining, and underwent caspase and JNK dependent cell death. Gene expression profiling of p21 knockdown cells that were treated with etoposide revealed altered expression of numerous genes, including upregulation of growth promoting genes and apoptotic genes, as expected. Indeed, apoptotic cell death appears to be the reasons why p21 knockdown cells lose viability following drug treatment. Finally, the authors demonstrate that p21^{-/-} mice show reduced liver fibrosis in response to CCl₄, owing to a reduced abundance of senescent cells in damaged liver.

General Comments:

While the study aims to identify mechanisms involved in maintaining cell viability during senescence, this reviewer is not convinced that senescence pathways had been activated by etoposide treatment. Drug treatment was performed for 48h, a timeframe that surely does not engage a full-blown senescence response. Etoposide is a topoisomerase inhibitor, which, therefore, requires cells to replicate DNA in order to develop double strand DNA breaks. A 48h drug treatment likely merely activates a DNA damage response that may or may not result in cellular senescence days or weeks following treatment. In addition, p21 has been demonstrated to be required for initiating a proliferative arrest after DNA damage, and it is therefore not surprising that eliminating p21 from the DNA damage response allows cells to continue to proliferate after drug treatment resulting in eventual apoptotic cell death.

In support of this reviewer's conclusion that drug treatment did not result in the activation of a senescence response are the authors' own data demonstrating that cells in oncogene induced senescence (which takes at least 7 days to develop) are insensitive to p21 ablation (Fig 1A).

Specific Comments:

If the authors wish to conclude that p21 maintains cell viability in senescence, and not simply after DNA damage, the authors must demonstrate that treatment with etoposide results in fully senescent cells that display SAβGal activity, enlarged/flat/spindle like morphology, SAHF, high levels of heterochromatin proteins, etc. at the 48h time point. Even if this can be accomplished, the authors should also confirm that long term senescence (weeks following drug treatment) induced by etoposide can also be destabilized by siRNA against p21.

Figure 1A: why are cells in OIS insensitive to p21 knockdown?

Figure S1: H1299 cells lack functional p53. Etoposide treatment of these cells has been demonstrated to cause apoptosis, not growth arrest. In addition, it has been demonstrated that p21 is not upregulated in these cells in response to etoposide treatment. Please explain these potential discrepancies.

Figure 3A: why is there no ser15 phosphorylation of p53 following drug treatment in siCtrl??

Figure 5A: It would be useful to verify that p21^{-/-} animals show reduced senescence in response to CCl₄ due to increased apoptosis in the tissue.

Referee #3:

In this paper, the authors investigated the contribution of pathways that regulate the cellular senescence to the resistance of these cells to cell death. More precisely, they studied how p21 regulates cell viability of senescent cells upon DNA damage.

They showed that p21 favors the viability of senescent cells in a cumulative effect, at least for DNA damage Induced Senescence (DIS) by regulating a large gene expression profile implicated in cell-cycle, TGF-β, DDR and focal adhesion signaling pathways. The knockdown of p21 in DIS cells diminished their viability; regulate gene expression, reinforced DDR (more γH2AX and 53BP1 foci, more pATM) induced by the resumption of DNA synthesis. They showed that the p21 increase of senescent cell viability is p53 independent; TNF-α release induced by p65 NF-κB activation dependent and pJNK dependent. Finally, they showed that p21 KO in vivo eliminates senescent cells in the liver and decreases collagen production that together limit liver fibrosis.

Globally, the paper is well written with clear and convincing figures and strong evidence for the conclusions. The paper brings the novel idea that senescent cell survival is p21 dependent and p53 independent, at least for DNA Damage Induced senescence. The conclusions are well discussed and

replaced in the domain context that is very competitive and has experienced great discoveries this year (Baker 2016, Chang, 2016, Yosef, 2016). However, some points remains unclear or are questionable.

Major comments:

1) The authors analyzed the impact of p21 KO in vivo in a model of DNA Damage Induced senescence in the liver. They clearly showed a decrease of senescent cell accumulation and a decrease of fibrotic lesions in the liver of p21 KO mice associated to the modulation of α -SMA, COL1A1 and TGF- β . They proposed that p21KO alleviate liver fibrosis not only by diminishing the quantity of senescent cells but also by diminishing TGF- β , COL1A1 and α -SMA. Thus, they analyzed the impact of p21 KD in vitro on DIS cells (Figure 5F, G and H). The effect of p21KD on the decrease expression of COL1A1 mRNA (5F) in DIS cells is clear and stable in the time. The diminution at the protein level for DIS cell upon p21 KD is also very clear. However, the authors claim that COL1 is also decreased in growing cell (5G) that is totally not convincing by the blot presented in 5G where we cannot see a clear modification. The author must provide convincing data on this point.

2) Since the authors have previously showed that they can rescue the p21KD phenotype in vitro with both pan-caspase inhibitor and JNK pathway inhibitor, it is mandatory in the context of this work to test whether the double treatment with these inhibitors can block the rescue of the liver fibrosis in p21KO mice.

Minor comments:

- The authors analyzed in figure 4 the impact of p53 and pRB in the phenotype of p21 dependent cell viability and showed that the mechanism is p53 independent. They showed that p21 KD favor NF κ B and JNK activation and trigger cell death through caspase and and JNK pathways.

a) In Figure 4c, they authors should add also the blot for total JNK

b) They showed in figure 4D that p21 KD induced an increase of TNF- secretion since 24h and reach a plateau at 48 hours whereas the increase of IL-8 secretion starts at 48h and is significantly increased at 72h. Since they showed in figure 1E that the mechanism is cumulative, it seems important to clarify in figure 4C and 4D, how many hours after siRNA transfection the experiment is performed ? Does the IL-8 secretion depend on TNF- α secretion? Is it a direct or indirect effect?

c) In figure 4E, does the authors could explain why the effect of p21 KD on p-JNK induction for DIS cells is much lower than in 4C for the DIS cells because the induction in 4E of pJNK is not really visible? Similar question for PARP cleavage.

d) In figure 4H, the statistic should be added to the figure like in 4G or 4I.

e) In figure 4I, it could be informative to also add the blots and not only the quantification to fully appreciate the total rescue of the cell viability obtain by the inhibition of caspase and JNK pathway.

Point-by-point response to the referees' comments

Yosef et al, EMBOJ-2016-95553

We thank the reviewers for their positive view on the novelty of our work and for the helpful suggestions.

Referee #1:

...The study touches on an area of general interest. The authors present some very interesting culture based work on a role of p21 in promoting survival of senescent cells by inhibiting cell cycle progression in the presence of DNA damage and thus by inhibiting a NFkB dependent TNFalpha induction. The in vivo work is also interesting but has some major flaws. It is well possible that p21 deletion improves the proliferative capacity of hepatocytes in mice exposed to repeated doses of CCl4. Activation of stellate cells in the context of chronic liver injury is triggered by the regenerative decline of hepatocytes. Therefore, it is well conceivable that improved hepatocyte proliferation in response to p21 deletion prevents the induction of stellate cell activation and the induction of stellate cell senescence rather than promoting the death of senescent stellate cells.

We appreciate that the reviewer notices the novelty and the general interest of our study and provided many helpful suggestions. We realize that the interpretation of the in vivo experiment might be more general owing the involvement of many different cell types in a disease in a context of the whole organ. In the original manuscript, both in results and discussion we already mentioned, in accordance with the reviewer note, that fibrosis is a complex disease and the effect of p21 knockdown might be mediated through several parallel mechanisms, including decreased stellate cell activation, decrease ECM secretion as well as the decrease senescent cell viability. Moreover, the original text in the results section clearly stated that "The significant effect of p21 on liver fibrosis cannot be fully explained, however, by the reduction in amount of senescent cells." This is in complete agreement with the above note of the reviewer. In principle, as the reviewer suggest, it is not possible to exclude that increase in proliferation of hepatocytes might contribute, together with other factors, to the decrease in activation of stellate cells in this model. We now added this note to the discussion (p. 17).

Specific points:

1) Fig.1. Just mentioning that "the cells had acquired the senescence phenotype" is not enough. SA-β-gal staining, proliferation curves and FACS plots of cell cycle distribution before and after etoposide washout, as well as other markers of senescence should be sufficiently provided in supplementary figure.

We performed the induction of senescence by standard protocols used by us and others (Narita et al, Cell 2006; Krizhanovsky et al, Cell 2008; Lujambio et al, Cell 2013; Biran et al, G&D 2015; Yosef et al, Nature Communications 2016, and many others). We now included Suppl. fig 1 which describes the procedure and all the parameters of the senescence phenotype in the cells we used, including sustainable cell cycle arrest assessed by different experimental approaches and increase in the expression of senescence markers on the protein level.

Minor comment: It is surprising that authors did not see the difference between control samples in graphs of Fig. 1A,B,C. Since PrestoBlue reagent, like any other similar reagents, estimates the amount of metabolically active cells, proliferating G and V groups must show higher values than senescent DIS and OIS. It seems that authors normalized to 100% all control samples, which is mathematically wrong to do since 100% in G group is not the same 100% in DIS group because of different normalization coefficient. So, either all values have to be normalized with one coefficient (like in Fig. 4G,H,I) or all groups have to be presented in separate plots, so nobody will compare controls between each other.

We agree with this suggestion and all groups are now presented in the separate plots.

2) Fig. S1. Experiment with p53-null H1299 cell line is not described in Materials & Methods. Transduction protocol, etoposide dose, period of treatment, acquiring time of senescent phenotype have to be explained. Similar to comment 1, markers of senescence have to be provided.

The details of the protocols are added to the Materials and Methods as suggested. Markers of senescence, including SA-beta-gal, p21 and the cell cycle profile by FACS are presented in the new Suppl. Figure 2. Of note, in no place in the manuscript we claimed that H1299 cells become senescent; we have shown that following DNA damage treatment the cells acquire cell cycle arrest and exhibit p21 up-regulation. In these conditions the cells respond to p21 knockdown with cell death, similarly to what we observed in senescent primary fibroblasts. The only point in this experiment is that the mechanism we study is relevant in multiple settings.

3) Fig. 3. From my point of view, this part of the study is rather not convincing and should be strengthened. More experiments should be performed to demonstrate DNA lesions and DNA-damage response after sip21. Comet assays, FACS analysis of sub-G1 percentage, WBs of involved proteins and FACS for quantitative estimation of γ H2AX and 53BP1 could be done.

We substantially extended our studies to analyze DNA damage and DNA damage response following p21 knockdown in senescent cells as suggested by the reviewer. The new experiments presented in the new Fig. 3 extensively analyzed DNA damage response following the p21 knockdown. We now show by both WB and immunofluorescence that multiple markers of DNA

damage response are increased in senescent cells following the p21 knockdown. We performed the experiments suggested by the reviewer and observed increase in DNA fragmentation in Comet assay and increase in sub-G1 population in FACS analysis following p21 knockdown (new Fig.3 H and L). These experiments firmly support our conclusion that p21 knockdown in senescent cells leads to increase in DNA damage and cell death.

a) Fig. 3A. Opposite to Fig. 4A, p53 protein in Fig. 3A is clearly stabilized after sip21 and demonstrate dramatic Ser-15 phosphorylation, which should be at least discussed, since authors claim that the decreased survival of DIS cells after sip21 is p53-independent. p53 phosphorylation is indeed dramatic, although amount of p-ATM foci increased from 2 to just 2.2 foci/cell, probably meaning another involved mechanism.

Following p21 knockdown we observe increase in the levels of p-p53 Ser-15 and some stabilization of total p53 which is evident on both Fig. 3A and Fig. 4A. Indeed the levels of total p53 seem to be stronger increased in Fig. 3A, which does not contradict at all our conclusions regarding the experiments in Figs. 3 and 4. Indeed, p53 is stabilized following p21 knockdown (Figs. 3A and 4A), but knockdown of p53 is not able to prevent cell death following p21 knockdown. Moreover, p53 knockdown itself induced cell death to almost same degree as p21 knockdown (Fig. 4A). We agree that this data can completely confirm that the cell death following p21 knockdown is p53-independent and we removed this claim from the revised manuscript. Regarding the increase in the markers of DNA damage and DNA damage itself we now added extensive experimental evidence as described in the previous answer (new Fig. 3) for significant increase in DNA damage and DNA damage response following p21 knockdown.

b) Fig. 3I. Since in this figure DAPI staining demonstrated such dramatic discrepancy with any other DAPI pictures, differences in sample preparations (time after sip21, etc.) should be explained in the figure legend. As mentioned above, additional methods for DNA lesion detection should be performed.

Minor comment: Typo in Fig. 3F - "53BP1", not "p53BP1".

We clarified the timing of the experiment in the text of the results and in the figure legend as suggested by the reviewer (24 hours for fig. 3B and 3I and 4 days for fig.3K). Indeed the cells in Fig 3K (former 3I) are presented 4 days after the p21 knockdown when about half of the cells is already dead and the others show signs of cell death and nuclear fragmentation as documented and as expected from the data presented in the manuscript. The cells in the new Fig 3I are 24 hours after the knockdown and thus most of the nuclei seen intact. These data is in complete agreement with the data presented in fig 1 and other panels of fig 3. It supports our conclusion that p21 knockdown leads to induction of DNA damage and cell death of DIS cells.

Additional experiments for detection of DNA damage were performed (Fig. 3) as requested and they are described above in the answer to point 3a.

The typo is corrected.

4) Fig. 4 raises a number of questions:

a) Fig. 4A. Authors claim "that p21 maintains the viability of senescent cells downstream of p53" and in p53-independent manner. This conclusion hardly be obtained from provided data, since sip53 does not reduce p21 amount in comparable with sip21 level but the survival of DIS cells after sip53 is similar to p21 knockdown. How authors explain such dramatic decrease of cell survival after sip53? Other targets of p53?

Indeed, survival of DIS cells is similar after knockdown of p21, p53 or both proteins together. There are several possible explanation of these results. We agree with the reviewer that it is not possible to firmly conclude that the effect is p53-independent and we now removed this statement from the manuscript. One possible explanation for decrease in viability following p53 knockdown, which is supported by the data, is that p53 knockdown leads to decrease in the levels of p21, which in turn initiated the molecular cascade involving JNK and caspase cleavage. This explanation is provided in the text. We can't exclude, of course, that other targets of p53 (and there are numerous targets known) are involved in this process.

b) Fig. 4B,E. There is a number of confusing and non-matching WB results. Why do pRb amount decreased after sip21 and, oppositely, the level of p21 dramatically increased after sipRb? Should be explained. Why there is a strong band of Cleaved-PARP in siCtrl sample in Fig. 4E, contradicting all other results in Fig.4E (is it the same experiment?)?

Molecular pathways explaining the changes in p21 following pRb knockdown and vice versa are described in the literature, and therefore are not surprising. On one hand, pRb knockdown can cause an increase in p16 and p21 in order to induce premature senescence in MSCs (Lin et al, 2014: <https://www.ncbi.nlm.nih.gov/pmc/articles/PMC4264040/>). We believe the same feedback loop exists when we knockdown pRb in DIS cells causing dramatic increase in p21. On the other hand, enhanced Mdm2 activity inhibits pRB function via ubiquitin-dependent degradation (Uchida et al, 2004: <http://emboj.embopress.org/content/24/1/160>). p53 stabilization following p21 knockdown observed in DIS cells may, therefore, lead to pRb degradation via the release of Mdm2. Owing that knockdown of pRb did not led to the rescue of cells following p21 knockdown we did not focused our studies on these pathways.

Following the next comment (regarding Fig 4CE) we repeated this experiments several times and our data firmly show that there is a substantial increase in PARP cleavage following sip21. In the original Fig. 4E the strong band of cleaved PARP was the full length PARP who technically didn't separated from the cleaved form. The repeated experiments show increase in PARP cleavage in DIS cells following p21 knockdown as shown at Fig. 4E. This increase is also accompanied with increase in cleavage of caspase 3 and death of these cells.

c) Fig. 4C,E. It is clear that p-p65 is increased after sip21 even without DIS. How authors would explain the difference in JNK and caspase activations in G and DIS groups? As well mechanism of increased JNK activation after p65 knockdown (Fig. 4E) should be explained. Importantly, total JNK level has to be provided as a control for phosphorylation blots (same for Fig. 4H).

The WB experiments were now performed few more times and the representative blots shown also for total JNK as suggested by the reviewer.

The requested explanations are now provided at the Discussion section. The main difference between the G and DIS group is activation of persistent DNA damage response in the DIS group before the introduction of the knockdown of p21. DIS cells also have increased TNF production (Fig. 2F). NF- κ B is able to induce p21 expression via TNF- α (Basile et al, 2003: <http://mcr.aacrjournals.org/content/molcanres/1/4/262.full.pdf>). Owing that p21 maintains the viability of DIS cells, its knockdown by siRNA can activate NF- κ B in order to provide a feedback and induce its upregulation both in G and in DIS cells. Low level of TNF- α levels will allow cell survival however high levels will induce cell death (Biton and Ashkenazi, 2011) all controlled by the DDR and ATM.

Caspases can be activated by DNA damage in parallel and not consequently to the NF κ B activation (Biton and Ashkenazi, 2011). Therefore, it is possible that while caspase cleavage is dependent on NF κ B activity, JNK activity is independent of this pathway. Indeed, the results presented in Fig. 4E demonstrate that knockdown of p65 alone does not affect JNK phosphorylation. Combined knockdown of p21 and p65 still prevents caspase cleavage, but does not rescue DIS cells from the cell death induced by p21 knockdown.

As the suggested by the reviewer in one of the comments below JNK phosphorylation is regulated by multiple pathways and can be regulated by p21 directly and therefore it is possible that in our system it is not regulated by p65 knockdown alone in Fig. 4E. This possibility is discussed in the revised manuscript.

d) Fig. 4D & 5F. Similar to comment 1, all values have to be normalized with only one coefficient or all groups have to be presented in separate histograms.

The groups are now presented in separate plots.

e) Fig. 4H,I. It is known that chemical inhibitors in general and MAPK inhibitors in particular usually demonstrate unspecific action (particularly for SP600125, see Tanemura et al., Curr. Enzym. Inhib. 2010; 6:26-33). That is why I would recommend to prove Fig. 4H,I experiments with anti-JNK siRNAs, importantly, since involvement of JNK pathway seems to be the key result of the study.

We now performed all the experiments that were originally performed with JNK inhibitor with siJNK. The data is significantly extended and shown in new Fig 5. The new data firmly supports our previous conclusions, namely that combination of JNK pathway inhibition with inhibition of caspase cleavage (siJNK with QVD in the new experiments) rescues the death following p21 knockdown.

f) Fig. 4I. One of the most important results of the paper misses WBs. Should be provided.

The WB is now provided as suggested. It is now new Fig 5C

Minor comment: The upper huge band in cleaved caspase 3 blots in Fig. 4G and Fig. S7 should be designated.

The nature of the band is well known and described in literature (Methot et al, JBC, 2004. "A Caspase Active Site Probe Reveals High Fractional Inhibition Needed to Block DNA Fragmentation"). The large subunit of fully processed caspase-3 migrated at 17 kDa (p17). Partially processed caspase-3 retains part (p19) or all (p20) of the pro-domain. Following QVD treatment there is complete abolishment of cc3 processing and we detect increase in p20 and disappearance of p17 and p19 as expected from the described QVD action. We realize that the two lower bands are the informative ones that represent caspase cleavage and therefore these two bands are now shown in all WBs presented.

5) Fig. 5. The main question here - does the decreased amount of senescent cells in p21^{-/-} livers show their excessive elimination (as stated in the abstract and introduction) or reduced formation? Principally, experiment with in vivo injections of sip21 could answer this question. Other potential players, JNK for example, could be validated as well.

We show that the amount of senescent cells in the p21 knockout livers is decreased. We also show in the revised manuscript that this is accompanied by an increase in apoptosis in the same areas. We agree with the point of the reviewer that we can't exclude reduction in the production of senescent cells as one of the causes for reduction in their presence. We now corrected the abstract and the introduction accordingly.

Induction of knockdown in vivo by siRNA is not well established technology and there is no indication that such siRNA treatment will reach stellate cells. Months of preliminary experiments will be needed in order to study this point before the requested experiment can be performed. Importantly, the JNK inhibition experiment was performed by other laboratory and is reported in the literature (Kluwe et al, Gastroenterology 2010). The results of this experiment show significant increase in pJNK in activated stellate cells and reduction of liver fibrosis by treatment with JNK inhibitor in the same mouse model we used in our study. Therefore, JNK inhibition

reduces liver fibrosis in mice by the direct effect on stellate cells. We now discuss these results in the manuscript.

Minor comment: a) Mouse strain has to be designated in Materials & Methods. b) The mechanism of collagen expression regulation by p21 should be discussed.

Mouse strain and its source are clearly stated at Materials and Methods.

Possible mechanism of collagen regulation includes mainly effect on TGF-beta pathway, which is the main regulator of collagen expression, and is now discussed as suggested.

General minor comments:

1) It is well known, and described in detail for example in the mentioned review of Abbas & Dutta, 2009, that p21 interacts with and thus directly regulates activity of JNK, upstream kinase ASK1 and procaspases. I would suggest that this level of regulation and its possible role in the observed effects should be considered and discussed.

This point is now considered in the Discussion section as suggested.

2) There are some non-correct citations. For example, Lanigan et al., 2011 at the page 11 (3rd line from the bottom) should be replaced to Biton & Ashkenazi, 2011, and Condiotti et al., 2014 at the page 18 (5th line) should be replaced to Choudhury et al., 2007. Authors should double-check all references.

The references were corrected and the accuracy of citation was re-checked.

3) In order to make the conclusions more comprehensible, authors could consider a diagram preparation, which would sum up their model of newly defined p21 actions.

The diagram is now included as Suppl. Fig. S9.

4) Materials & Methods. Transduction protocols should be described with more details or references should be provided. The same is for transfection procedure, at least transfection reagent should be mentioned. PD number of primary cells has to be stated. Phosphorylation sites for all relevant antibodies have to be named.

All the requested information above is now included in the Materials and Methods section.

5) Statistical analysis. Instead of SEM, which just represents the mathematical preciseness of mean value calculation, standard deviation SD generally more preferable in biology, since

represent data variability, which is more informative. Authors could for example look at Tom L. Croat. Med. J. 2004;45:361-70.

This is a general practice to use SEM for presentation of the data in biological experiments. While in some circumstances SD might be more preferable, this is not the general practice in the literature in biology in general. For example in the latest issue of EMBO Journal (volume 36, issue 4) there are 9 research articles and only one of them uses SD in the presentation of the data. All other articles, that present the results as bar graphs, use SEM like we did. In any case the presentation of the data does not change the statistical significance of the tests and does not affect the validity of our conclusions.

6) Immunoblotting analysis. Generally, there is a good practice of providing uncropped and unprocessed immunoblot pictures in supplementary figure. I would strongly recommend to do it.

The un-cropped row WBs of the main results are now included as suppl Fig. S10.

Referee #2:

...General Comments:

While the study aims to identify mechanisms involved in maintaining cell viability during senescence, this reviewer is not convinced that senescence pathways had been activated by etoposide treatment. Drug treatment was performed for 48h, a timeframe that surely does not engage a full blown senescence response. Etoposide is a topoisomerase inhibitor which, therefore, requires cells to replicate DNA in order to develop double strand DNA breaks. A 48h drug treatment likely merely activates a DNA damage response that may or may not result in cellular senescence days or weeks following treatment. In addition, p21 has been demonstrated to be required for initiating a proliferative arrest after DNA damage, and it is therefore not surprising that eliminating p21 from the DNA damage response allows cells to continue to proliferate after drug treatment resulting in eventual apoptotic cell death.

In support of this reviewer's conclusion that drug treatment did not result in the activation of a senescence response are the authors own data demonstrating that cells in oncogene induced senescence (which takes at least 7 days to develop) are insensitive to p21 ablation (Fig 1A).

We agree with the reviewer that the phenotype takes 7 days to develop in either DIS or OIS settings. We performed the induction of senescence by standard protocols used by us and others many times over more than 10 years (Narita et al, Cell 2006; Krizhanovsky et al, Cell 2008; Lujambio et al, Cell 2013; Biran et al, G&D 2015; Yosef et al, Nature Communications 2016). We appreciate that the reviewer points out that the timeline of induction of senescence was not described sufficiently clear. In the revised manuscript we included the timeline of the

induction as suppl. Fig. 1a. This shows that we considered the cells senescent 7 days after the etoposide treatment.

In the revised manuscript we included Suppl. fig 1 which describes multiple characteristics of the senescence phenotype in the cells we used, including sustainable cell cycle arrest assessed by different experimental approaches and increase in the expression of senescence markers on the protein level. Moreover, microarray analysis performed on these cells and described in Fig. 2 revealed that DIS cells exhibit the expression profile that contains all the characteristics of senescence and which is similar to the expression profiles published to other DIS cells.

Based on the above we firmly believe that we used DIS cells in our experiments as stated in the text of the manuscript.

Specific Comments:

If the authors wish to conclude that p21 maintains cell viability in senescence, and not simply after DNA damage, the authors must demonstrate that treatment with etoposide results in fully senescent cells that display SAbGal activity, enlarged/flat/spindle like morphology, SAHF, high levels of heterochromatin proteins, etc. at the 48h time point. Even if this can be accomplished, the authors should also confirm that long term senescence (weeks following drug treatment) induced by etoposide can also be destabilized by siRNA against p21.

As discussed above we now included the characteristics of DIS cells we used in suppl. Fig. 1. To address the point regarding the sustainability of the effect of p21 knockdown in cells that were senescent for some time we performed a p21 knockdown experiment in cells that are senescent for 3 weeks. The results of this experiment are presented in Suppl fig. 1F and they show continuous decrease in the viability of senescent cells in these conditions following the p21 knockdown. Therefore, presence of senescent cells in culture for long time does not make them insensitive to the p21 knockdown.

Figure 1A: why are cells in OIS insensitive to p21 knockdown?

There is more than one possible reason why OIS are insensitive to p21 knockdown. From one side HrasV12 is potent oncogene that activates variety of signaling pathways, some of which strongly support cell survival (for example Bonni et al, Science 1999). From the other side OIS cells might up-regulate p16 to much higher levels than DIS cells and p16 could easier substitute for p21 in this system as both have similar molecular function or might inhibit JNK directly (Bu Young Choi et al, Nature Structural Molecular Biology 2005). Therefore, there are several possible explanations for the insensitivity of OIS cells to p21 knockdown. We included the discussion of these possibilities in the Discussion section of the revised manuscript.

Figure S1: H1299 cells lack functional p53. Etoposide treatment of these cells has been demonstrated to cause apoptosis, not growth arrest. In addition, it has been demonstrated that p21 is not upregulated in these cells in response to etoposide treatment. Please explain these potential discrepancies.

Etoposide treatment, like any other DNA damage, can cause either apoptosis or cell cycle arrest depending on the dose. The dose we used induced cell cycle arrest in H1299 cells (Suppl Fig 2). However, this does not contradict in any way that higher dose of etoposide would be able to induce apoptosis in this cells. The same is true for any other cell in culture, including primary fibroblasts.

Indeed, p21 is one of the most known transcriptional targets of p53. However, this does not mean that p53 is the only regulator of p21. Regulation of p21 expression is rather complex. Our data firmly shows that in our experimental settings p21 is upregulated in H1299 cells, like in many other cell lines lacking p53, in response to DNA damage. The variety of the pathways that can regulate p21 expression independent of p53 are extensively discussed (Abbas and Dutta Nature Reviews Cancer 2009) and are beyond the scope of our work. We used the H1299 cells, as well as MEFs, to serve as example that the p21 knockdown induced cell death is not strictly limited to the senescence human fibroblasts.

Figure 3A: why is there no ser15 phosphorylation of p53 following drug treatment in siCtrl??

The increase in phosphorylation of p53 happens also in siCtrl DIS cells comparing to siCtrl growing cells. Higher exposure of the same blot shown below demonstrates this point. The blot presented in Fig. 3A shows exposure that allows seeing the most prominent increase in phospho-p53 on this blot and this is in sip21 in DIS cells, without overexposing the blot. One can notice a presence of a band in siCtrl DIS sample and absence of the one in siCtrl growing sample.

Figure 5A: It would be useful to verify that p21^{-/-} animals show reduced senescence in response to CCl₄ due to increased apoptosis in the tissue.

To evaluate apoptosis in WT and p21^{-/-} mice in response to CCl₄ as suggested by the reviewer we performed staining of the liver sections on CCl₄ treated and control mice from both

genotypes with a marker of apoptosis cleaved caspase 3. The results are included in the revised manuscript as Fig.6B and they indicate increase in apoptosis in p21^{-/-} livers as expected.

Referee #3:

....Globally, the paper is well written with clear and convincing figures and strong evidences for the conclusions. The paper bring the novel idea that senescent cell survival is p21 dependent and p53 independent, at least for DNA Damage Induced senescence. The conclusions are well discussed and placed in the domain context that is very competitive and has experienced great discoveries this year (Baker 2016, Chang, 2016, Yosef,2016). However, some points remains unclear or are questionable.

Major comments:

1) The authors analyzed the impact of p21 KO in vivo in a model of DNA Damage Induced senescence in the liver. They clearly showed a decrease of senescent cell accumulation and a decrease of fibrotic lesions in the liver of p21 KO mice associated to the modulation of α -SMA, COL1A1 and TGF- β . They proposed that p21KO alleviate liver fibrosis not only by diminishing the quantity of senescent cells but also by diminishing TGF- β , COL1A1 and α -SMA. Thus, they analyzed the impact of p21 KD in vitro on DIS cells (Figure 5F, G and H). The effect of p21KD on the decrease expression of COL1A1 mRNA (5F) in DIS cells is clear and stable in the time. The diminution at the protein level for DIS cell upon p21 KD is also very clear. However, the authors claim that COL1 is also decreased in growing cell (5G) that is totally not convincing by the blot presented in 5G where we cannot see a clear modification. The author must provide convincing data on this point.

We identified decrease in the expression of Col1 in growing cells for the first time during the analysis of the microarray data (Fig. 2), which was surprising for us as well. We decided that even the result is surprising and less understood we would follow it up in the context of this study. QPCR experiments confirmed that Col1 expression is reduced in growing cells following p21 knockdown (Fig 2F). Based on this observation we then followed up in fig 5 with analysis of Col1 expression on the protein level. We now repeated this experiment several times and the representative blot (Fig 6H) clearly demonstrates reduction in the Col1 levels in sip21 vs siCtrl in both senescent and growing cells. Owing the surprise of the result in growing cells we went further and analyses total collagen secretion from these cells using Sirius Red (the same dye used to analyze ECM presence in liver sections). The results (Fig 6 I,J) show clear and significant reduction in overall collagen production and deposition by growing cells. Altogether, our data firmly supports that p21 regulates collage expression in both growing and senescent cells.

2) Since the authors have previously showed that they can rescue the p21KD phenotype in vitro with both pan-caspase inhibitor and JNK pathway inhibitor, it is mandatory in the context of this work to test whether the double treatment with these inhibitors can block the rescue of the liver fibrosis in p21KO mice.

The experiment using JNK inhibition in the same mouse model we used was performed by other laboratory and is reported in the literature (Kluwe et al, Gastroenterology 2010). The results of this experiment show significant increase in pJNK in activated stellate cells and reduction of liver fibrosis by treatment with JNK inhibitor. Therefore, JNK inhibition reduces liver fibrosis in mice by the direct effect on stellate cells. We now discuss these results in the manuscript. We think that further in vivo experiments with different inhibitors and their combinations are beyond the scope of our study.

Minor comments:

- The authors analyzed in figure 4 the impact of p53 and pRB in the phenotype of p21 dependent cell viability and showed that the mechanism is p53 independent. They showed that p21 KD favor NFkB and JNK activation and trigger cell death through caspase and and JNK pathways.

a) In Figure 4c, they authors should add also the blot for total JNK

The blot is provided in the revised figure 4 as suggested.

b) They showed in figure 4D that p21 KD induced an increase of TNF- secretion since 24h and reach a plateau at 48 hours whereas the increase of IL-8 secretion starts at 48h and is significantly increased at 72h. Since they showed in figure 1E that the mechanism is cumulative, it seems important to clarify in figure 4C and 4D, how many hours after siRNA transfection the experiment is performed ? Does the IL-8 secretion depend on TNF-a secretion? Is it a direct or indirect effect?

The timing of the cell collection is now clearly stated in the figure legend as suggested. In the figure 4D the timing after siRNA is actually indicated on the x axis. The assay in 4C and 4E was done 4 days after siRNA washout. It is evident that the increase in the expression of TNF happens earlier than cell death and secretion of IL-8 follows much later, when cell death already occurs. In general, TNF has the ability to induce massive inflammatory response in tissue culture. It was shown (Biton and Ashkenazi, 2011) that RIP1 kinase promotes JNK-mediated induction of IL-8 production to alert other cells of the damage and recruits FADD to activate Caspase-8 and trigger programmed cell death. Therefore, we think that increase in the levels of IL-8 is a secondary effect and it is parallel of the cell death we observed following p21 knockdown.

c) In figure 4E, does the authors could explain why the effect of p21 KD on p-JNK induction for DIS cells is much lower than in 4C for the DIS cells because the induction in 4E of pJNK is not really visible? Similar question for PARP cleavage.

Following the suggestion to include JNK blots we repeated all the experiments and performed new WB experiments for pJNK and JNK. The new results show that pJNK is substantially increased in DIS cells comparing to growing cells (Fig 4C). The levels of pJNK are further increased by combined knockdown of p21 and p65, but not either of the two alone (Fig. 4E).

The results of PARP cleavage as presented in the original fig 4C (same blot new fig 4C as well) show substantial cleavage of PARP following p21 knockdown. The experiment presented in fig 4E was repeated several more times and the results firmly show increase in PARP cleavage following p21 knockdown. These results are supported by the observed cleavage of caspase 3 and the death of DIS cells in these conditions.

Altogether the results presented in the new Fig 3 C and E are incomplete agreement with each other and with the rest of the data presented in Figures 3, 4 and 5.

d) In figure 4H, the statistic should be added to the figure like in 4G or 4I.

The statistics is added as suggested – now the panel is Fig. 5B.

e) In figure 4I, it could be informative to also add the blots and not only the quantification to fully appreciate the total rescue of the cell viability obtain by the inhibition of caspase and JNK pathway.

These experiments were further extended with siJNK as suggested by reviewers and WB is provided for all experiments and presented in the new Fig 5.

Thank you for submitting the revised version of your manuscript. It has now been seen by the three original referees, whose comments are enclosed below.

As you will see, the first referee remains more critical on the study than the two others, however we decided - in light of the strong support of the latter - to give you the opportunity to revise your manuscript to address this referee's points. Thus, I would like to invite you to submit a final revised version of the manuscript using the link enclosed below, addressing the reviewer's comments.

Both referee #2 and referee #3 find that their concerns have been sufficiently addressed and are broadly in favour of publication. Referee #1 states, that your claims on the *in vivo* part of the study are not sufficiently well supported by the current data, and is asking you to consolidate your findings on apoptosis in p21 KO stellate cells. This referee also points out that there is a need for you to revise your western blot data and provide complementary source data files, as well as to address a number of other minor concerns regarding controls, formatting and data representation.

Thus, I ask you to revise your manuscript regarding the points raised by referee #1 and evaluate, whether you would be able to add supportive complementary data as indicated, or, alternatively, relativise your statements and introduce caveats where appropriate. Please note, that while these points are well taken, taking into account the positive comments of referee #2 and referee #3, we have decided that pending a satisfactory revision, we would go ahead with acceptance of this manuscript as soon as possible.

Regarding the source data requested earlier, we realised that you included additional western blot data in Supplemental Figure 10. As to our image analysis, there appear to be some potential aberrations in Figures 3A and 5A, which are likely the result of image compression. Nevertheless, we would ask for a full set of source data for these figures. As to micrographs in Figure 3D - there appears to be a text artifact behind the assembled panel. Thus, we ask you to provide full sets of source data for these figures as high-resolution individual images, in order to clarify the issues. Please note that these source data are required to proceed with processing of your manuscript.

 REFEREE REPORTS

Referee #1:

In general, the manuscript has improved and the authors addressed most of my comments.

I still find the *in vivo* part rather weak. The authors should include some stellate cell/apoptosis co-staining to demonstrate that apoptosis affects this cell type in p21 KO mice. As senescence is restricted to activated stellate cells this would provide direct evidence that p21-deletion leads to elimination of senescent cells *in vivo*. Alternatively, this may not be the case and p21 deletion improves hepatocyte proliferation in the context of CCl₄ induced DNA damage. In that case apoptosis may affect some hepatocytes continuing to proliferate in the presence of DNA damage. According to this scenario, the reduction in senescent cells may just be indicative of better hepatocyte proliferation reserve. These two experiments should be conducted, as the entire *in vivo* part is otherwise rather vague and unclear.

I also have remaining concerns regarding the Western Blot data: Suppl Fig. 10 lacks full blot images of many panels:

- Fig. 3A - pCHK2, Tubulin
- Fig. 4A - p53,
- Fig. 4B - p21, pRb, Tubulin
- Fig. 4C - p21, cleaved PARP, Cleaved - CASP3, Tubulin
- Fig. 4E - completely missing

Fig. 5C - p21, Tubulin,
Fig. 5E - p21, Actin

Fig. 6H - lacking

S1E - lacking
S2D - lacking
S8A - lacking

Tubulin in Fig. 3A looks identical to 4C

Minor points:

The authors should use γ H2AX instead of p- γ H2AX as γ H2AX is already referring to phosphorylated H2AX.

The authors often use the term "dramatic" in order to describe their results. I find this term a bit overstatement and not very scientifically.

Almost all of the bar graphs in Figure 4 lack p-values, e.g. 4A,B,D,E - The authors, however, draw conclusions like "it thus seems that p21 knockdown can induce death of senescent cells in the absence of p53 (Fig. 4A)" - however, the difference between sip53 and sip21 may not be significant and thus the conclusion is possibly not validated by the data.

Figure 3G - the y-axis is unclear. What does normalized frequency mean? Why is that an algorithmic scale without any numbers? Are the differences between sip21 and siCtrl significant?

Referee #2:

The authors have adequately addressed my concerns and questions.

Referee #3:

The revised version is satisfactory.

2nd Revision - authors' response

04 May 2017

Second revision

Point-by-point response to the referees' comments

Yosef et al, EMBOJ-2016-95553

Referee #1:

In general, the manuscript has improved and the authors addressed most of my comments.

I still find the in vivo part rather weak. The authors should include some stellate cell/apoptosis co-staining to demonstrate that apoptosis affects this cell type in p21 KO mice. As senescence is restricted to activated stellate cells this would provide direct evidence that p21-deletion leads to elimination of senescent cells in vivo. Alternatively, this may not be the case and p21 deletion improves hepatocyte proliferation in the context of CCl₄ induced DNA damage. In that case apoptosis may affect some hepatocytes continuing to proliferate in the presence of DNA damage. According to this scenario, the reduction in senescent cells may just be indicative of better hepatocyte proliferation reserve. These two experiments should be conducted, as the entire in vivo part is otherwise rather vague and unclear.

The staining for the apoptosis marker cleaved caspase 3 presented on the figure 6B clearly shows that apoptosis is not increased randomly in p21 knockout mice. It is increased along the fibrotic scar areas where stellate cells are present. Therefore, apoptosis does not seem to occur at the areas where hepatocytes are present and rather occurs in the areas where stellate cells are present, thus strongly supporting our conclusions. This being said we cannot exclude the possibility that the proliferation of hepatocytes might be affected in the p21 knockout mice. We have already included this possibility in the discussion. In order to accommodate the reviewers' comment we now extended this point and included explicit possibility for the alternative explanation suggested by the reviewer.

The corrected statement in the revised manuscript, including the reviewer's comment, is "Alternatively p21 deletion might be able to improve hepatocyte proliferation in the context of CCl₄ induced DNA damage. In that case apoptosis may affect some hepatocytes continuing to proliferate in the presence of DNA damage."

I also have remaining concerns regarding the Western Blot data: Suppl Fig. 10 lacks full blot images of many panels:

The source files are now provided to all the figures mentioned below as separate files according to the editor's instructions.

Fig. 3A - pCHK2, Tubulin

Fig. 4A - p53,
Fig. 4B - p21, pRb, Tubulin
Fig. 4C - p21, cleaved PARP, Cleaved - CASP3, Tubulin
Fig. 4E - completely missing

Fig. 5C - p21, Tubulin,
Fig. 5E - p21, Actin

Fig. 6H - lacking

S1E - lacking
S2D - lacking
S8A - lacking

Tubulin in Fig. 3A looks identical to 4C

Experiments in 4C were repeated. The results of these experiments are similar to our previous results and firmly support our conclusions. The results, including the new tubulin blot, are presented in new Fig 4C. There are no changes in the layout of this panel or in the observed changes in protein levels, comparing to the results presented earlier in this panel.

Minor points:

The authors should use γ H2AX instead of p- γ H2AX as γ H2AX is already referring to phosphorylated H2AX.

The manuscript and the figures were corrected as suggested.

The authors often use the term "dramatic" in order to describe their results. I find this term a bit overstatement and not very scientifically.

The term was replaced as suggested

Almost all of the bar graphs in Figure 4 lack p-values, e.g. 4A,B,D,E - The authors, however, draw conclusions like "it thus seems that p21 knockdown can induce death of senescent cells in the absence of p53 (Fig. 4A)" - however, the difference between sip53 and sip21 may not be significant and thus the conclusion is possibly not validated by the data.

The statistical analysis was added to the panels on Fig. 4 as requested. Indeed, the difference between sip21 and sip53 is not significant. In fact we never relate to this small difference as significant and the above conclusion was drawn based on the functional effects

of the knockdowns on the cell viability, when sip53 is not able to abolish the effect of sip21 on the viability. Based on these experiments it is clear that knockdown of p21 indeed induces cell death in senescent cells in absence of p53. In order to further weaken the statement we corrected the conclusion to "it thus seems that p21 knockdown might induce death of senescent cells in the absence of p53 (Fig. 4A)". This statement better reflects the possibility that there are alternative explanations that can be suggested to our results.

Figure 3G - the y-axis is unclear. What does normalized frequency mean? Why is that an algorithmic scale without any numbers? Are the differences between sip21 and siCtrl significant?

The definition of the normalized frequency added to the methods section. The numbers were added to the y-axis as suggested by the reviewer. The difference between sip21 and siCtrl are significant as identified by the statistical analysis. A figure presenting the means, the SEMs and the statistical analysis is added as appendix Fig S3.

Reviewers 2,3

We are happy to see that reviewers 2 and 3 have no more comments and find our revised manuscript suitable for publication. We appreciate their input for improvement of our manuscript.

3rd Editorial Decision

08 May 2017

Thank you for submitting the revised version of your manuscript. I have now evaluated your revised manuscript and discussed your response to the remaining concerns of referee #1 in the editorial team. From these considerations we have concluded that the concerns have been sufficiently addressed.

Thus, I am pleased to inform you that your manuscript has been accepted for publication in the EMBO Journal.

Corresponding Author Name: Valery Krizhanovsky

Manuscript Number: EMBOJ-2016-95553R